# Linking skeletal muscle aging with osteoporosis by lamin A/C deficiency

**Lei Xiong**[1,2,3☯]**, Kai Zhao**[1,3☯]**, Yu Cao**[3]**, Hao-Han Guo**[1]**, Jin-Xiu Pan**[1,2]**, Xiao Yang**[3]**, Xiao Ren**[1]**, Lin Mei**[1,2,3]**, Wen-Cheng Xiong**[1,2,3]*

**1** Department of Neurosciences, School of Medicine, Case Western Reserve University, Cleveland, Ohio, United States of America, **2** Louis Stoke VA Medical Center, Cleveland, Ohio, United States of America, **3** Department of Neuroscience and Regenerative Medicine, Medical College of Georgia, Augusta University, Augusta, Georgia, United States of America

☯ These authors contributed equally to this work.
* Wen-Cheng.Xiong@case.edu

## Abstract

The nuclear lamina protein lamin A/C is a key component of the nuclear envelope. Mutations in the lamin A/C gene (*LMNA*) are identified in patients with various types of laminopathy-containing diseases, which have features of accelerated aging and osteoporosis. However, the underlying mechanisms for laminopathy-associated osteoporosis remain largely unclear. Here, we provide evidence that loss of lamin A/C in skeletal muscles, but not osteoblast (OB)-lineage cells, results in not only muscle aging–like deficit but also trabecular bone loss, a feature of osteoporosis. The latter is due in large part to elevated bone resorption. Further cellular studies show an increase of osteoclast (OC) differentiation in cocultures of bone marrow macrophages/monocytes (BMMs) and OBs after treatment with the conditioned medium (CM) from lamin A/C–deficient muscle cells. Antibody array screening analysis of the CM proteins identifies interleukin (IL)-6, whose expression is markedly increased in lamin A/C–deficient muscles. Inhibition of IL-6 by its blocking antibody in BMM-OB cocultures diminishes the increase of osteoclastogenesis. Knockout (KO) of IL-6 in muscle lamin A/C–KO mice diminishes the deficits in trabecular bone mass but not muscle. Further mechanistic studies reveal an elevation of cellular senescence marked by senescence-associated beta-galactosidase (SA-β-gal), p16$^{Ink4a}$, and p53 in lamin A/C–deficient muscles and C2C12 muscle cells, and the p16$^{Ink4a}$ may induce senescence-associated secretory phenotype (SASP) and IL-6 expression. Taken together, these results suggest a critical role for skeletal muscle lamin A/C to prevent cellular senescence, IL-6 expression, hyperosteoclastogenesis, and trabecular bone loss, uncovering a pathological mechanism underlying the link between muscle aging/senescence and osteoporosis.

## Introduction

Nuclear lamins, originally identified as lamins A, B, and C, are major components of the inner nuclear membrane, the so-called nuclear lamina [1]. They interact with other inner nuclear

**Data Availability Statement:** All relevant data are within the paper and its supporting information files.

**Funding:** This work was supported in part by the US National Institutes of Health (https://www.nih.

gov) (AG051773 to W-CX, AG051510 to LM) and Department of Veterans Affair (https://www.va. gov) (BX000838 to W-CX). The funders had no role in study design, data collection and analysis, decision to publish, or preparation of the manuscript.

**Competing interests:** The authors have declared that no competing interests exist.

**Abbreviations:** μCT, microcomputer tomographic; ALP, alkaline phosphatase; BFR, bone formation rate; BMD, bone mineral density; BMM, bone marrow macrophage/monocyte; BMP, bone morphogenetic protein; BMSC, bone marrow stromal cell; BS, bone surface; BubR1, Bub1-related kinase; BV/TV, bone volume over total volume; CAAX, cysteine-aliphatic-aliphatic-any amino acid; CDM1A, cardiomyopathy, dilated, 1A; cko, conditional knockout; CM, conditioned medium; CMT2, Charcot–Marie–Tooth disorder, type 2; CMT2B1, Charcot–Marie–Tooth disease, type 2B1; Cre, cyclization recombination enzyme; DCM, dilated cardiomyopathy; DMD, Duchenne muscular dystrophy; Dmd$^{mdx}$, X chromosome–linked muscular dystrophy mice; EDMD2, autosomal dominant Emery–Dreifuss muscular dystrophy; EDMD3, autosomal recessive Emery–Dreifuss muscular dystrophy; FGF, fibroblast growth factor; FPLD2, familial partial lipodystrophy, Dunnigan type; HGPS, Hutchinson–Gilford progeria syndrome; HHS-S, Slovenian type heart–hand syndrome; HSA, human alpha-skeletal actin; IκBα, inhibitor of nuclear factor kappa B; IL, interleukin; IL-1ra, IL-1 receptor antagonist; IL-6R, IL-6 receptor; KD, knockdown; KO, knockout; LGMD1B, limb-girdle muscular dystrophy type 1B; L-CMD, *LMNA*-related congenital muscular dystrophy; LMNA, lamin A/C gene; Lmna$^{f/f}$, floxed Lmna mice; Lmna$^{HG/HG}$, gene-targeted mice with HGPS mutation; Lmna$^{HSA}$-cko, skeletal muscle–specific Lmna-cko mice; Lmna$^{Ocn}$-cko, OB-selective Lmna–conditional knockout mice; MADA, mandibuloacral dysplasia with type A lipodystrophy; MADB, mandibuloacral dysplasia type B; MAR, mineral apposition rate; M-CSF, macrophage colony-stimulating factor; MIG, monokine induced by gamma interferon; MNC, multinucleated cell; mo, months old; MS, mineral surface; NF-κB, nuclear factor kappa-light-chain-enhancer of activated B cells; NHEJ, nonhomologous end joining; OB, osteoblast; OC, osteoclast; Ocn, osteocalcin; OPG, osteoprotegerin; PTHrP, parathyroid hormone–related peptide; PYD, pyridinoline; RANK, receptor activator of NF-κB; RANKL, receptor activator of NF-κB ligand; RANTES, regulated on activation, normal T cell expressed and secreted; RB, retinoblastoma; RD, restrictive dermopathy; ROS,

membrane proteins on the interior of the nuclear envelope and provide not only structural support for the cell nucleus but also transcriptional regulation of the chromosome DNAs [1,2]. A-type lamins are produced by alternatively spliced products of the lamin A/C gene (*LMNA*), which include lamin A and C [3]. Lamin C does not require posttranslational modification to localize to the inner nuclear membrane. However, lamin A is generated in a precursor form, called pre–lamin A, which undergoes a series of posttranslational modifications to become mature lamin A. These modifications include farnesylation at a cysteine-aliphatic-aliphatic-any amino acid (CAAX) motif of the carboxyl-terminal cysteine, endoproteolytic release of the terminal amino acids, carboxymethylation of the accessible farnesylcysteine, and removal of the final 15 carboxyl-terminal residues by zinc metallopeptidase STE24 (ZMPSTE24) [4–7].

Mutations in genes encoding lamins result in a heterogeneous group of human genetic disorders known as laminopathies [8,9], which include muscular dystrophy diseases (such as autosomal dominant Emery–Dreifuss muscular dystrophy [EDMD2]; autosomal recessive Emery–Dreifuss muscular dystrophy [EDMD3]; limb-girdle muscular dystrophy type 1B [LGMD1B]; *LMNA*-related congenital muscular dystrophy [L-CMD]; and cardiomyopathy, dilated, 1A [CDM1A]), lipodystrophy (e.g., familial partial lipodystrophy, Dunnigan type [FPLD2], and mandibuloacral dysplasia with type A lipodystrophy [MADA]), peripheral nerve disorders (e.g., Charcot–Marie–Tooth disease, type 2B1 [CMT2B1]), and premature aging syndromes (e.g., Hutchinson–Gilford progeria syndrome [HGPS] and restrictive dermopathy [RD]) [10–21]. Although it remains unclear exactly how mutant *LMNA* causes laminopathies, it is believed that both gain of pre–lamin A's toxic functions and loss of mature lamin A/C's physiological functions contribute to the disease pathology [22]. Patients with premature aging syndromes, such as HGPS and RD, often have mutations in *LMNA* or *ZMPSTE24* gene, which result in the accumulation of toxic levels of pre–lamin A in cells [13,15,17,21]. Studies on patients and lamin A/C–deficient mouse models indicate that loss of function of lamin A/C proteins also develops some of the laminopathies, including muscular dystrophy, dilated cardiomyopathy (DCM), and Charcot–Marie–Tooth disorder, type 2 (CMT2) [14,23,24]. Although these observations demonstrate the important function of lamin A/C to prevent laminopathies, it remains unclear exactly how the loss of lamin A/C affects multiple tissues and promotes various types of laminopathies.

Laminopathies are phenotypically diverse, affecting not only skeletal muscles but also cardiac muscles, adipose tissue, peripheral nerves, and bone [8]. It is noteworthy that although laminopathies have diverse symptoms, many of them share overlapping features of skeletal muscle dystrophy and osteoporosis [8,22]. Osteoporosis, an age-associated skeletal degenerative disorder, is characterized by a decrease of bone mass, micro–architectural deterioration of bone tissue, and an increase of hip fracture rate [25,26]. It is associated with several laminopathies, including HGPS, MADA, mandibuloacral dysplasia type B (MADB), L-CMD, and Slovenian type heart–hand syndrome (HHS-S) [22]. Whereas most studies have been focused on lamin A/C's function in preventing skeletal muscle dystrophy, few reports are available regarding the relationship between laminopathies and osteoporosis and whether and how lamin A/C regulates bone mass homeostasis and prevents osteoporosis.

Here, we provide evidence that loss of lamin A/C in skeletal muscles results in not only muscular dystrophy–like phenotypes but also osteoporotic deficit. The bone loss deficit is likely due to increased osteoclastogenesis and osteoclast (OC)-mediated bone resorption. Cellular studies demonstrate that the conditioned medium (CM) of Lmna-KO muscle cells promotes osteoclastogenesis by increasing receptor activator of nuclear factor κB ligand (RANKL)/osteoprotegerin (OPG) expression in osteoblasts (OBs). Antibody array screening of the secreted proteins in the CMs of Lmna-KO C2C12 muscle cells identified interleukin

reactive oxygen species; SA-β-gal, senescence-associated beta-galactosidase; SASP, senescence-associated secretory phenotype; shRNA, short hairpin RNA; sIL-6R, soluble IL-6R; Tb.N, trabecular bone number; Tb.Sp, trabecular bone space; TGF, transforming growth factor; TRAP, tartrate-resistant acid phosphatase; ZMPSTE24, zinc metallopeptidase STE24; WT, wild-type.

(IL)-6, a proinflammatory cytokine that is markedly increased in the CMs of Lmna-KO muscle cells. Inhibition of IL-6 by its blocking antibody or genetic knocking out diminishes the increase of osteoclastogenesis and trabecular bone loss driven by Lmna-KO muscles. Further mechanical studies suggest that the increased IL-6 expression in Lmna-KO muscle cells is due in large part to the increase of p16$^{Ink4a}$, a marker and a driver of cellular senescence. Taken together, these results suggest a critical role for muscle lamin A/C to regulate osteoclastogenesis, revealing a link between skeletal muscle dystrophy and osteoporosis in patients with laminopathies.

## Results

### Trabecular bone loss in skeletal muscle–selective, but not OB-selective, Lmna-KO mice

To understand how osteoporosis is linked with laminopathies, we first generated OB-selective Lmna–conditional KO (cko) mice (Lmna$^{Ocn}$-cko), as OBs are the key bone formation cells, which express high levels of lamin A/C (S1A and S1B Fig). Lmna$^{Ocn}$-cko mice were generated by crossing the floxed Lmna mice (Lmna$^{f/f}$) with osteocalcin (Ocn)-cyclization recombination enzyme (Cre) (S1A Fig), which expresses Cre largely in OB-lineage cells [27,28]. Western blot analysis showed a reduction of lamin A/C protein in bone marrow stromal cells (BMSCs), but not bone marrow macrophages/monocytes (BMMs), derived from Lmna$^{Ocn}$-cko mice (S1B Fig). Characterization of phenotypes in Lmna$^{Ocn}$-cko mice at 3 months old (mo), compared with same-aged littermate control mice (Lmna$^{f/f}$), showed little change in their body weight, body shape, subcutaneous fat thickness, and bone mass (S2A–S2L Fig), suggesting little, if any, role of lamin A/C in OB-lineage cells in bone homeostasis at young adult age.

We then generated skeletal muscle–specific Lmna-cko mice (Lmna$^{HSA}$-cko) by crossing Lmna$^{f/f}$ with human alpha-skeletal actin (HSA)-Cre transgenic mice (S1A Fig). The selective reduction of lamin A/C in skeletal muscles, but not in other tissues (e.g., heart, liver, brain, and thyroid), in Lmna$^{HSA}$-cko mice was observed by western blot analysis (S1C Fig). Little to no change of lamin A/C expression levels was detected in BMSCs and BMMs derived from Lmna$^{HSA}$-cko mice (S1D Fig). In contrast from Lmna$^{Ocn}$-cko mice, Lmna$^{HSA}$-cko mice had less body weight (Fig 1A and 1B) and developed kyphosis in their spines starting at the age of 2–3 mo (Fig 1A). Microcomputer tomographic (μCT) images showed an unsmooth spine bone morphology in 6-mo Lmna$^{HSA}$-cko mice (Fig 1C). The subcutaneous adipocyte size and fat thickness in Lmna$^{HSA}$-cko mice were also significantly reduced in an age-dependent manner (Fig 1D–1I). These phenotypes resembled to a certain degree the features of laminopathy (e.g., progeroid)-like pathology. As kyphosis is often associated with weak back muscles or spine bones, we performed HE staining analysis to examine both tissues. Lmna$^{HSA}$-cko mice (at age of 3 mo) had not only fewer muscle fibers (Fig 1J and 1K) but also lower spine bone mass (Fig 1J and 1L), suggesting that loss of lamin A/C in skeletal muscles is likely to affect both muscle and bone tissues.

We further examined phenotypes in the gastrocnemius muscles of Lmna$^{HSA}$-cko mice (3 mo). As shown in Fig 2A, the cross-sectional area of the muscles was significantly lower in Lmna$^{HSA}$-cko mice than that of control mice, Lmna$^{f/f}$ mice (Fig 2B and 2C), or HSA-Cre mice (S3A and S3B Fig). Interestingly, the mutant muscle fibers had central distributed or mispositioned nuclei, a feature of muscle degeneration/dystrophy (Fig 2A and 2C) (S3A and S3C Fig). Additionally, Lmna$^{HSA}$-cko mice showed decreased twitch and tetanic forces (Fig 2D–2F), supporting the view for muscle weakness/degeneration.

We then further examined bone deficit in long bones (e.g., femur) of 3-mo Lmna$^{HSA}$-cko mice by the μCT and HE analyses. Decreases in trabecular, but not cortical, bone volumes over

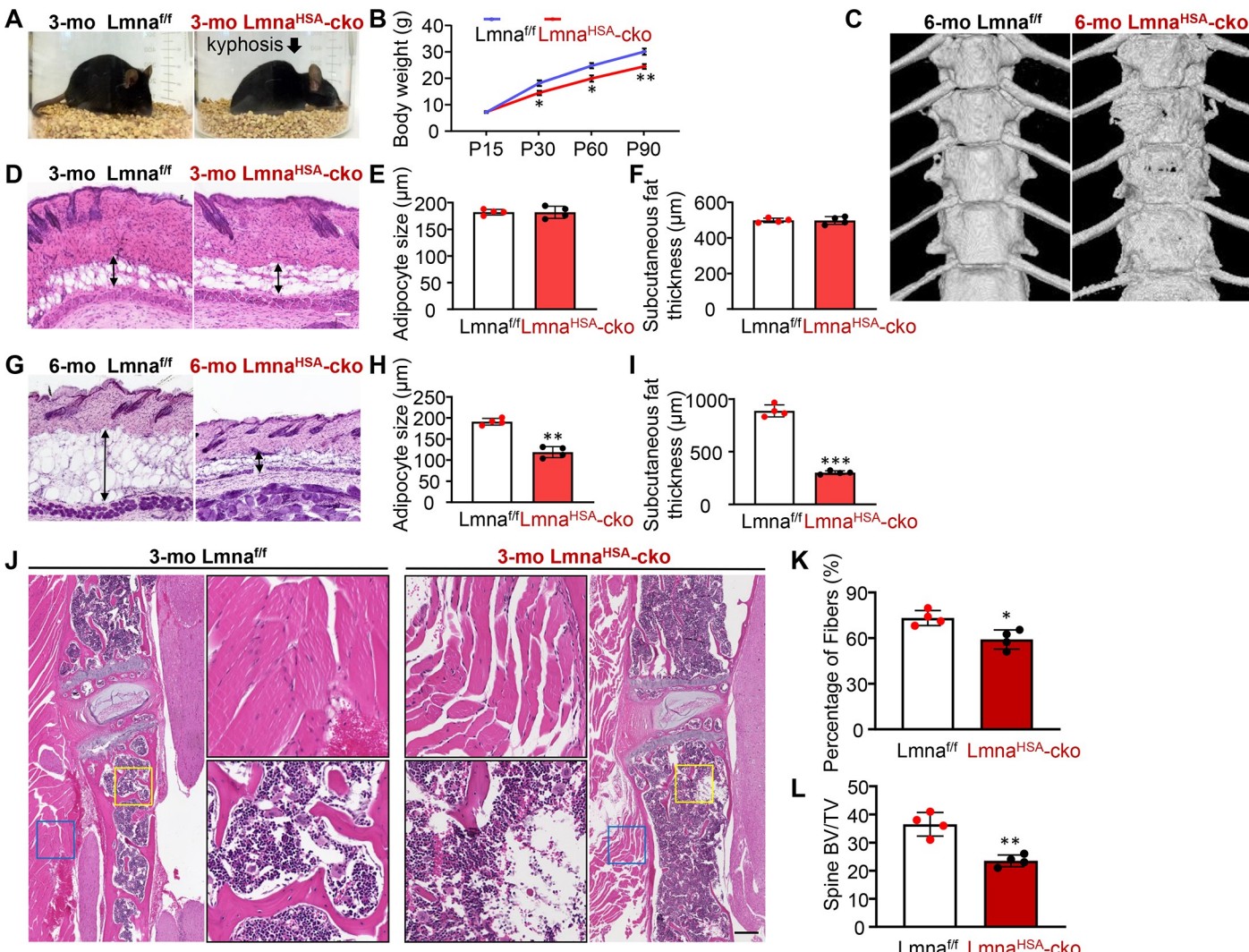

**Fig 1. Laminopathy-like phenotypes in muscle-selective Lmna-KO mice.** (A) Kyphotic-like phenotype in 3-mo Lmna$^{HSA}$-cko mice. Shown are representative images of mice with the indicated genotype. The control mice are littermates, Lmna$^{f/f}$. (B) Age-dependent decrease of body weight in Lmna$^{HSA}$-cko mice. $^*P < 0.05$, $^{**}P < 0.01$. (C) Representative µCT images of thoracic spines of 6-mo Lmna$^{f/f}$ and Lmna$^{HSA}$-cko mice. (D and G) Representative images of HE-stained skin sections from Lmna$^{f/f}$ and Lmna$^{HSA}$-cko mice at of 3 mo (D) and 6 mo (G). Scale bar, 200 µm. (E,F,H,I) Quantification of adipocyte size and subcutaneous fat thickness in Lmna$^{f/f}$ and Lmna$^{HSA}$-cko mice at 3 mo (E,F) and 6 mo (H,I). The values shown are means ± SD ($n = 4$ animals per genotype). $^{**}P < 0.01$, $^{***}P < 0.001$. (J) Representative images of spine muscle fibers and bone by HE staining analysis. Scale bar, 200 µm. (K,L) Quantification analyses of data in (J). The percentage of muscle fibers (K) and the spine trabecular BV/TV (L) were presented. The values shown are means ± SD ($n = 4$ animals per genotype). $^*P < 0.05$, $^{**}P < 0.01$. The underlying data for this figure can be found in S1 Data. µCT, microcomputer tomographic; BV/TV, bone volume over total volume; cko, conditional knockout; HSA, human alpha-skeletal actin; KO, knockout; Lmna, lamin A/C gene; Lmna$^{f/f}$, floxed Lmna mice; Lmna$^{HSA}$-cko, skeletal muscle–specific Lmna-cko mice; mo, months old; P, postnatal day.

total volumes (BV/TV), trabecular bone number (Tb.N), trabecular bone thickness, and increase in trabecular bone space (Tb.Sp) were all detected in Lmna$^{HSA}$-cko mice as compared with those of littermate controls (Fig 2G and 2H) (S3D and S3E Fig). Together, these results demonstrate trabecular bone loss not only in spine bones but also in long bones in young adult Lmna$^{HSA}$-cko mice, suggesting an important role of muscle lamin A/C in regulating trabecular bone mass homeostasis.

Notice that the muscle deficits were undetectable in Lmna$^{HSA}$-cko mice at 1 mo (S4A–S4F Fig), and these mice also had normal body shape without kyphosis. However, their trabecular

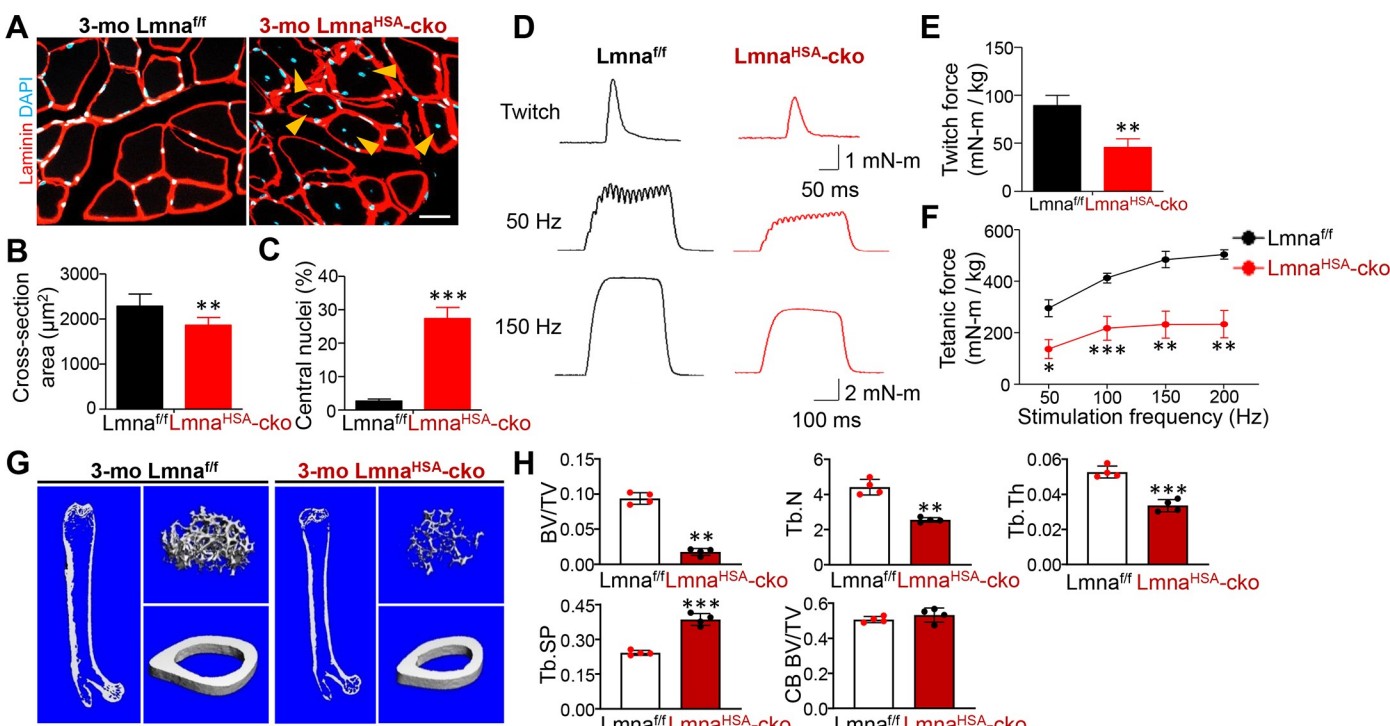

**Fig 2. Decreases in muscle size, muscle force, and TB mass in 3-mo Lmna^HSA^-cko mice.** (A) Representative images of gastrocnemius cross sections. Scale bar, 20 μm. (B,C) Quantification analyses of cross-section area and central nuclei distribution. The values shown are means ± SD ($n = 5$ animals per genotype). **$P < 0.01$, ***$P < 0.001$. (D) Representative twitch curves and tetanic curves at stimulation frequencies 50 and 150 Hz by muscle stimulation in Lmna^HSA^-cko mice. (E,F) Quantification analyses of twitch force and tetanic force. The values shown are means ± SD ($n = 4$ animals per genotype). *$P < 0.05$, **$P < 0.01$, ***$P < 0.001$. (G) Representative μCT 3D images of femurs from 3-mo Lmna^f/f^ and Lmna^HSA^-cko littermates. (H) Quantification analyses of TB BV/TV, Tb.N, TB.Th, Tb.Sp, and CB BV/TV by direct model of μCT analysis. Data are shown as means ± SD ($n = 4$ male mice of each genotype, examined blindly). **$P < 0.01$, ***$P < 0.001$, significant difference by two-way ANOVA. The underlying data for this figure can be found in S1 Data. μCT, microcomputer tomographic; BV/TV, bone volume over total volume; CB, cortical bone; cko, conditional knockout; HSA, human alpha-skeletal actin; KO, knockout; Lmna, lamin A/C gene; Lmna^f/f^, floxed Lmna mice; Lmna^HSA^-cko, skeletal muscle–specific Lmna-cko mice; mo, months old; TB, trabecular bone; Tb.N, TB number; Tb.Sp, TB space; TB.Th, TB thickness.

bone mass was significantly reduced at 1 mo (S4G and S4H Fig), suggesting that the bone deficit appeared to be an earlier-onset phenotype.

## Normal bone formation, but increased bone resorption, in Lmna^HSA^-cko mice

The decreased trabecular bone mass in Lmna^HSA^-cko mice may result from reduced bone formation and/or enhanced bone resorption. We thus first examined bone formation in Lmna^HSA^-cko mice. Measurement of serum levels of Ocn, a marker of bone formation, showed comparable levels in 3-mo Lmna^HSA^-cko mice with that of control mice, Lmna^f/f^ (Fig 3A), suggesting little to no alteration in bone formation. This view was further verified by two injections of the fluorochrome label calcein green, separated by a 10-day interval, to littermates of Lmna^HSA^-cko and Lmna^f/f^ mice. As shown in Fig 3B–3E, the mineral apposition rate (MAR), mineral surface (MS)/bone surface (BS), and bone formation rate (BFR), which were assessed in non-decalcified histological sections of femurs and tibia, were all comparable in Lmna^HSA^-cko mice to those of controls. Together, these results demonstrate normal bone formation in Lmna^HSA^-cko mice.

We next examined bone resorption in Lmna^HSA^-cko mice. Serum levels of pyridinoline (PYD) cross-link, a marker for bone resorption, were measured. At both 1 mo and 3 mo,

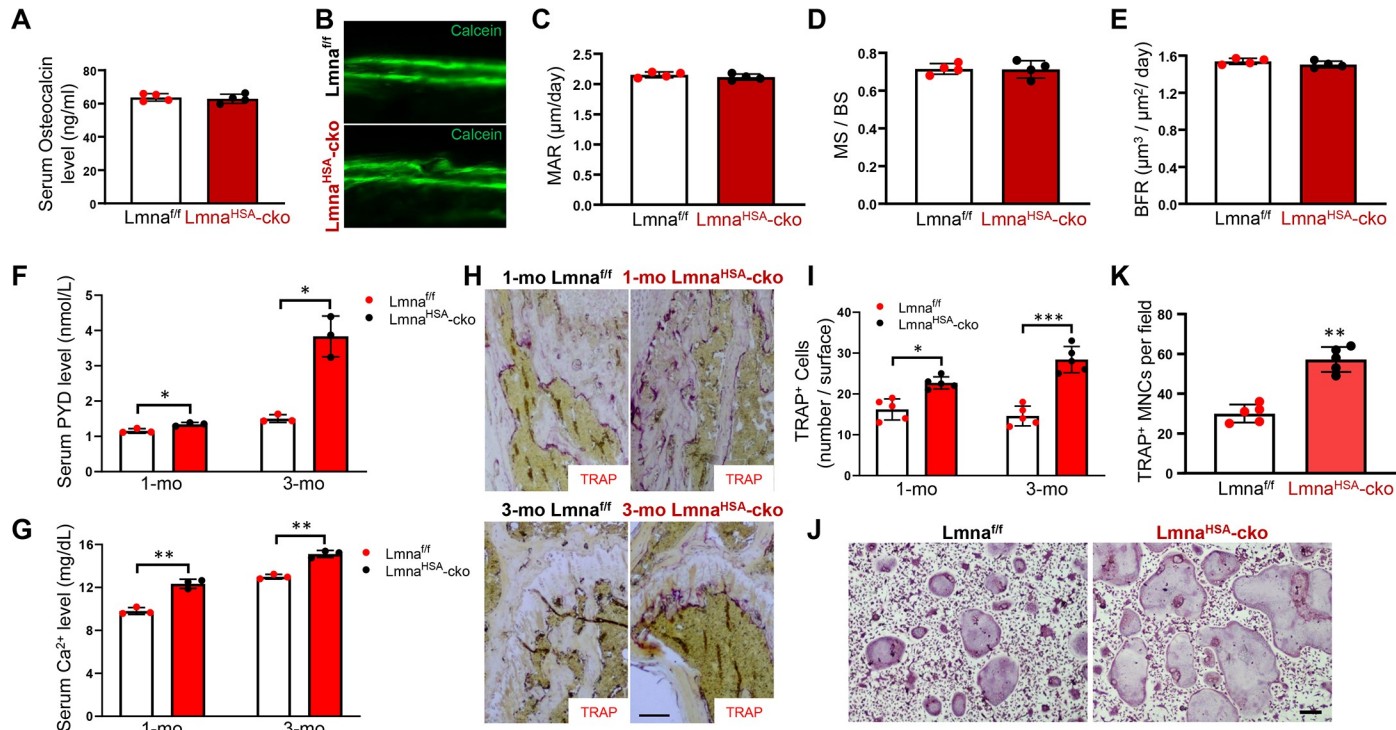

**Fig 3. No change in bone formation, but an increase in bone resorption, in Lmna$^{HSA}$-cko mice.** (A) Serum Ocn levels in 3-mo Lmna$^{f/f}$ and Lmna$^{HSA}$-cko mice. The serum Ocn levels were measured by ELISAs. The values shown are means ± SD from four males per genotype. (B) Representative images of histologic sections showing calcein labeling of endocortical bone in femur mid-diaphysis of Lmna$^{HSA}$-cko and Lmna$^{f/f}$ mice. (C-E) Ec.MAR (C), MS/BS (D), and Ec.BFR (E) are presented. The values shown are means ± SD from four males per genotype. (F) ELISA analysis of serum PYD levels. The values presented are means ± SD ($n$ = 3). $^{*}P < 0.05$, significant difference. (G) Colorimetric analysis of serum calcium levels. The values presented are means ± SD ($n$ = 3). $^{**}P < 0.01$, significant difference. (H) TRAP staining analysis of femur sections from 1-mo and 3-mo Lmna$^{f/f}$ and Lmna$^{HSA}$-cko mice. Scale bar, 100 μm. (I) Quantification analysis as means ± SD ($n$ = 5 femur samples for each group). $^{*}P < 0.05$, $^{***}P < 0.001$, significant difference. (J) TRAP staining analysis of cultured OCs derived from 3-mo Lmna$^{f/f}$ and Lmna$^{HSA}$-cko mice. Cells were treated with 100 ng/ml RANKL for 7 days. (K) Quantitative data of TRAP$^+$ MNCs (more than three nuclei) per randomly selected visual field. Scale bar, 100 μm. Data shown are means ± SD from five different cultures. $^{**}P < 0.01$. The underlying data for this figure can be found in S1 Data. BFR, bone formation rate; BS, bone surface; cko, conditional knockout; Ec.BFR, endocortical bone formation rate; Ec.MAR, endocortical mineral apposition rate; HSA, human alpha-skeletal actin; Lmna, lamin A/C gene; Lmna$^{f/f}$, floxed Lmna mice; Lmna$^{HSA}$-cko, skeletal muscle–specific Lmna-cko mice; MAR, mineral apposition rate; MNC, multinucleated cell; mo, months old; MS, mineral surface; NF-κB, nuclear factor kappa-light-chain-enhancer of activated B cells; OC, osteoclast; Ocn, osteocalcin; PYD, pyridinoline; RANKL, receptor activator of NF-κB ligand; TRAP, tartrate-resistant acid phosphatase.

serum PYD levels were significantly increased in Lmna$^{HSA}$-cko mice as compared with that of their littermate controls (Fig 3F). In addition, serum levels of calcium, another marker of bone resorption, were also increased in 1-mo and 3-mo Lmna$^{HSA}$-cko mice (Fig 3G). Increased bone resorption could be due to an increase in OC numbers or osteoclastogenesis. We thus examined the number of tartrate-resistant acid phosphatase positive (TRAP$^+$) OCs in femurs. Indeed, more TRAP$^+$ OCs were detected in Lmna$^{HSA}$-cko mice than that of littermate controls at 1 and 3 mo (Fig 3H and 3I). Furthermore, in vitro experiments showed that BMMs derived from 3-mo Lmna$^{HSA}$-cko mice had enhanced OC differentiation ability in response to RANKL (Fig 3J and 3K). Taken together, these results suggest that lamin A/C in skeletal muscles plays an important role in suppressing OC genesis.

## Increased in vitro osteoclastogenesis by CMs of Lmna-KO myotubes

To understand how Lmna KO in muscles regulates OC genesis, we generated an Lmna-KO C2C12 cell line (a mouse myoblast cell line) by using CRISPR-Cas9 system. Sequencing analysis confirmed the early terminal codon generated by nonhomologous end joining (NHEJ) in

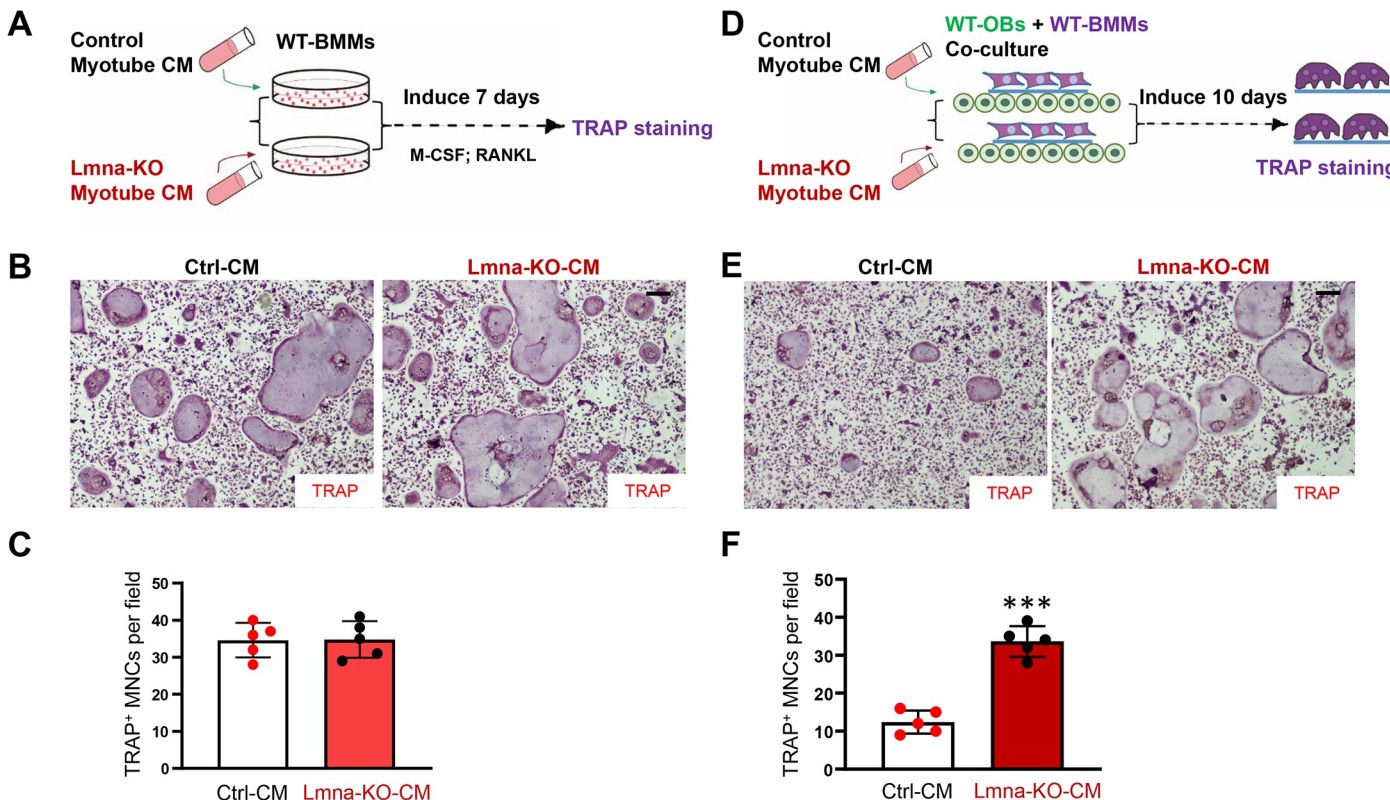

**Fig 4. Increased osteoclast differentiation in coculture system in the CM of Lmna-KO myotubes.** (A) The experimental treatment strategy. WT BMMs were treated with CMs of Ctrl or Lmna-KO myotubes. (B) The representative images of TRAP staining. Scale bar, 100 μm. (C) The quantitative data of TRAP+ MNCs. (D) Experimental strategy. WT OBs and WT BMMs were treated with CM of Ctrl or Lmna-KO myotubes for 10 days. (E) Representative images of the cultures treated with CM for 10 days. Scale bar, 100 μm. (F) The quantitative analyses of TRAP+ MNCs per field. The values shown are means ± SD from five different cultures. ***$P < 0.001$, significant difference from the Ctrl. The underlying data for this figure can be found in S1 Data. BMM, bone marrow macrophage/monocyte; CM, conditioned medium; Ctrl, control; KO, knockout; Lmna, lamin A/C gene; M-CSF, macrophage colony-stimulating factor; MNC, multinucleated cell; NF-κB, nuclear factor kappa-light-chain-enhancer of activated B cells; OB, osteoblast; RANKL, receptor activator of NF-κB ligand; TRAP, tartrate-resistant acid phosphatase; WT, wild-type.

cell clones 1–1 and 1–3 (S5A Fig). Western blot analysis verified Lmna KO in C2C12 cells (S5B Fig). The control and Lmna-KO C2C12 cells were induced into myotubes, and their CMs were collected. We then examined these CMs' effect on OC differentiation in the following in vitro assays. First, wild-type (WT) BMMs were incubated with the CM of control or Lmna-KO myotubes in the presence of macrophage colony-stimulating factor (M-CSF) and RANKL for 7 days (Fig 4A). TRAP staining analysis showed comparable numbers of TRAP+ cells in Lmna-KO CM–treated groups to those of control CM (Fig 4B and 4C), suggesting little to no direct role for in vitro OC differentiation by the Lmna-KO CM. Second, coverslips containing WT BMMs were cocultured with WT OBs in the presence of control or Lmna-KO CM for 10 days as illustrated in Fig 4D. The cells on coverslips were then subjected to TRAP staining. As shown in Fig 4E and 4F, the numbers of TRAP+ MNCs were higher in cocultures with Lmna-KO CM than that of control CM. These results suggest that the CM of Lmna-KO myotubes promotes OC genesis in a manner dependent on OB cells.

OBs are known to regulate OC differentiation through the expression and release of RANKL/OPG [29]. We thus examined these CMs' effect on OB differentiation and OB-derived RANKL/OPG expression. BMSCs were induced for OB genesis in the presence of the CM of control or Lmna-KO myotubes (S6A Fig). Alkaline phosphatase (ALP) staining analysis

showed comparable ALP$^+$ (OB) cells with or without the Lmna-KO CM (S6B and S6C Fig). However, the real-time PCR analysis showed higher RANKL expression but slightly lower OPG expression in OB cultures exposed to the Lmna-KO CM (S6D and S6E Fig). The ratio of RANKL over OPG was thus markedly elevated in OBs exposed to the mutant CMs (S6F Fig). In aggregate, the CM of Lmna-KO myotubes promotes OC differentiation, likely by its up-regulation of RANKL/OPG ratio in OBs, suggesting that Lmna-KO in muscles may regulate RANKL/OPG expression in OBs by secreted factors.

## Increased IL-6 in CMs of Lmna-KO myotubes and in the serum of Lmna$^{HSA}$-cko mice

To identify factors in the CM of Lmna-KO myotubes that may promote osteoclastogenesis, we carried out an antibody array screening analysis. In this array, 40 candidate factors, including various types of cytokines, were screened because many cytokines are implicated in osteoclastogenesis [30,31]. As shown in Fig 5A and 5B, four factors, IL-6, IL-16, monokine induced by

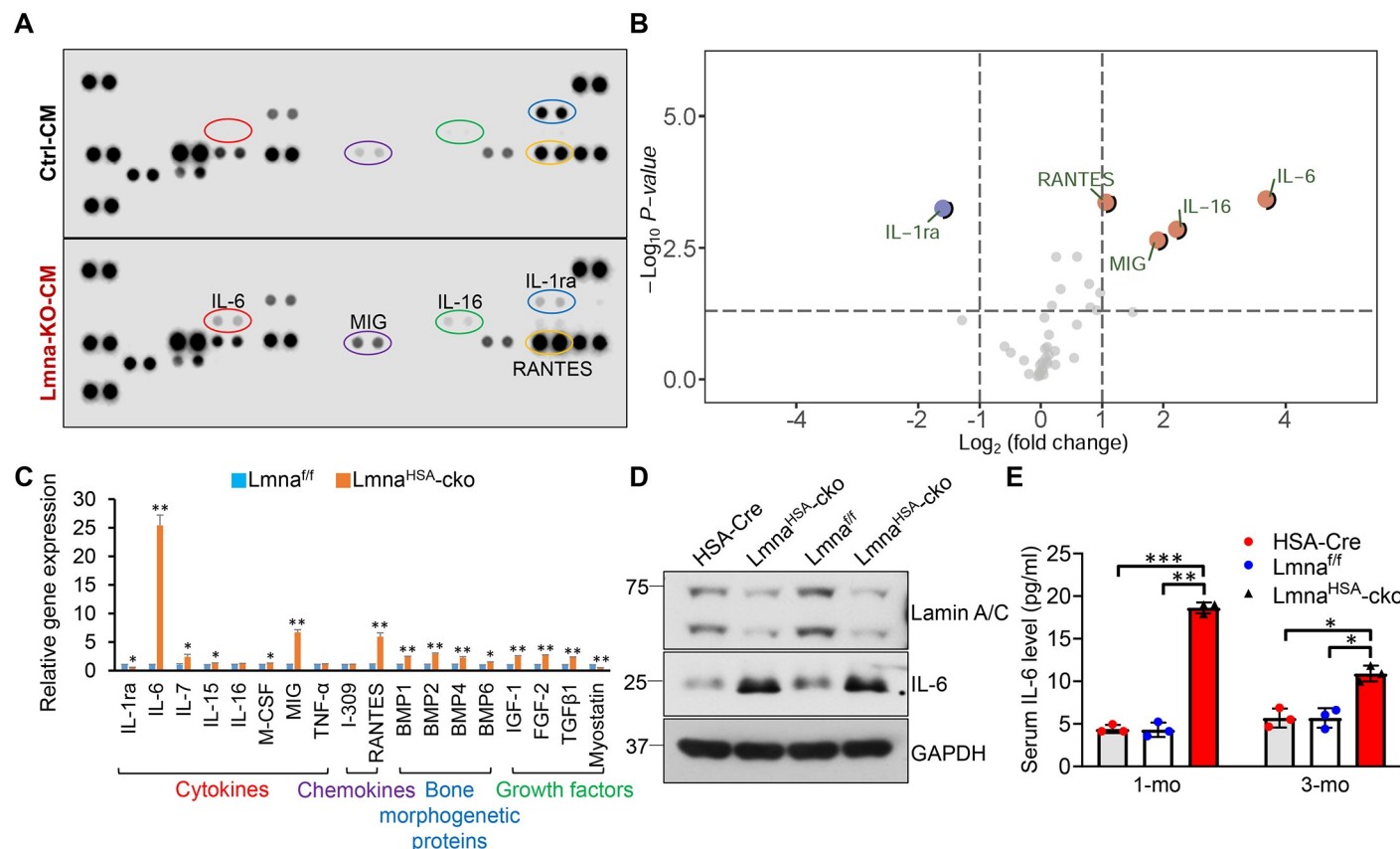

**Fig 5. Increased expression levels of IL-6 in Lmna-KO myotubes and in muscles and serum samples of Lmna$^{HSA}$-cko mice.** (A) Proteome profile of Mouse Cytokine Array of CM of Ctrl and Lmna-KO myotubes. (B) Volcano plot comparing Log2 (fold change). Up-regulated proteins were marked in red, and down-regulated proteins were indicated in blue ($P < 0.05$). (C) Real-time PCR analysis of gene expression in muscles of 1-mo Lmna$^{f/f}$ and Lmna$^{HSA}$-cko mice. $^*P < 0.05$, $^{**}P < 0.01$, significant difference. (D) Western blot analysis of IL-6 expression in muscles of 1-mo HSA-Cre, Lmna$^{f/f}$, and Lmna$^{HSA}$-cko mice. GAPDH was used as the loading Ctrl. (E) ELISA analysis of serum IL-6 level in 1-mo and 3-mo HSA-Cre, Lmna$^{f/f}$, and Lmna$^{HSA}$-cko mice. $^*P < 0.05$, $^{**}P < 0.01$, $^{***}P < 0.001$, significant difference. The underlying data for this figure can be found in S1 Data. BMP, bone morphogenetic protein; cko, conditional knockout; CM, conditioned medium; Cre, cyclization recombination enzyme; Ctrl, control; GAPDH, glyceraldehyde-3-phosphate hydrogenase; HSA, human alpha-skeletal actin; IL, interleukin; KO, knockout; Lmna, lamin A/C gene; Lmna$^{f/f}$, floxed Lmna mice; Lmna$^{HSA}$-cko, skeletal muscle–specific Lmna-cko mice; M-CSF, macrophage colony-stimulating factor; MIG, monokine induced by gamma interferon; mo, months old; RANTES, regulated on activation, normal T cell expressed and secreted; TGF, transforming growth factor; TNF, tumor necrosis factor.

gamma interferon (MIG), and regulated on activation, normal T cell expressed and secreted (RANTES), were significantly higher, whereas interleukin-1 receptor antagonist (IL-1ra) was lower, in the CM of Lmna-KO myotubes. The increased expressions of IL-6, MIG, and RANTES in Lmna-KO muscles were also detected by real-time PCR analysis (Fig 5C). In addition, the expressions of many growth factors (e.g., bone morphogenetic protein [BMP] 2, fibroblast growth factor [FGF]-2, transforming growth factor [TGF] β1) were elevated in Lmna-KO muscles (Fig 5C). Among the increased cytokines or growth factors, IL-6 stands out for the following reasons. First, it had the highest fold of increase in the mutant CMs and in the mutant muscles (Fig 5A–5D). Second, it is involved in osteoclastogenesis based on literature reports [32,33]. We thus further measured serum levels of IL-6 in 1-mo and 3-mo mice by ELISA. Indeed, the serum levels of IL-6 were elevated in Lmna^HSA-cko mice at both ages (Fig 5E).

To determine whether Lmna regulates IL-6 expression in OB-lineage cells as it does in muscle cells, we examined IL-6 expression and IL-6 levels in the CM in Lmna-KO and control BMSCs by real-time PCR and ELISA analyses, respectively. Both IL-6 mRNAs and protein levels in the CM were increased in Lmna-KO BMSCs (S7D and S7E Fig). However, the fold of IL-6 increase was less than that in Lmna-KO muscle cells (S7C Fig). In addition, the serum levels of IL-6 in 3-mo Lmna^Ocn-cko mice were comparable to that of control mice (S7B Fig). These results suggest that whereas lamin A/C in OB-lineage cells, as with that in muscle cells, is necessary to suppress IL-6 expression, its impact on IL-6 expression in OB-lineage cells appeared to be less than that in muscle cells. These results also implicate IL-6 as a potential factor underlying the increase of osteoclastogenesis by Lmna-KO muscles or CMs.

## Diminished in vitro osteoclastogenesis by IL-6 blocking antibodies

To determine whether IL-6 is the factor for the increased in vitro osteoclastogenesis by the CM of Lmna-KO myotubes, we took advantage of the IL-6 blocking antibody. To this end, myotube CM of control, Lmna-KO, and Lmna-KO plus IL-6 blocking antibody were added to the OB-BMM coculture dishes as illustrated in Fig 6A. At 10 days after the cocultures, cells on the coverslips were subjected to TRAP staining; and OBs in the culture plates were collected, and their total RNAs were subjected for real-time PCR. As shown in Fig 6B and 6C, IL-6 blocking antibody indeed abolished the increase of osteoclastogenesis in the coculture assays. The IL-6 blocking antibody also significantly decreased the ratio of RANKL/OPG in OBs exposed to the CM of Lmna-KO myotubes (Fig 6D). These results support the notion that IL-6 in the CM of mutant myotubes is a key factor for the increase of RANKL/OPG expression and in vitro osteoclastogenesis.

## Attenuated trabecular bone loss in Lmna^HSA-cko mice by IL-6 KO

To determine whether IL-6 contributes to the increase of osteoclastogenesis and bone resorption in Lmna^HSA-cko mice, we crossed Lmna^f/f; HSA-Cre mice with IL-6–null (IL-6–KO) mice to get the double mutant mice (S8A Fig). As expected, the resulting IL-6–KO; Lmna^HSA-cko double mutant mice showed undetectable serum levels of IL-6 that were similar to those of IL-6–KO mice (S8B Fig). These double mutant mice, as well as the single mutants, were then subjected to μCT analysis. Compared with Lmna^f/f control, IL-6–KO, or Lmna^HSA-cko mice, the IL-6–KO; Lmna^HSA-cko double mutant mice also exhibited kyphosis in their spines at 3 mo, which was similar to Lmna^HSA-cko mice (S8A Fig). The cross-sectional area of gastrocnemius muscles in IL-6–KO; Lmna^HSA-cko mice was similar to that of Lmna^HSA-cko mice (S9A and S9B Fig); and the nuclei in the double mutant mice remained centrally distributed (S9C Fig). These results suggest little effect on the muscle deficits in Lmna^HSA-cko mice by IL-6 KO.

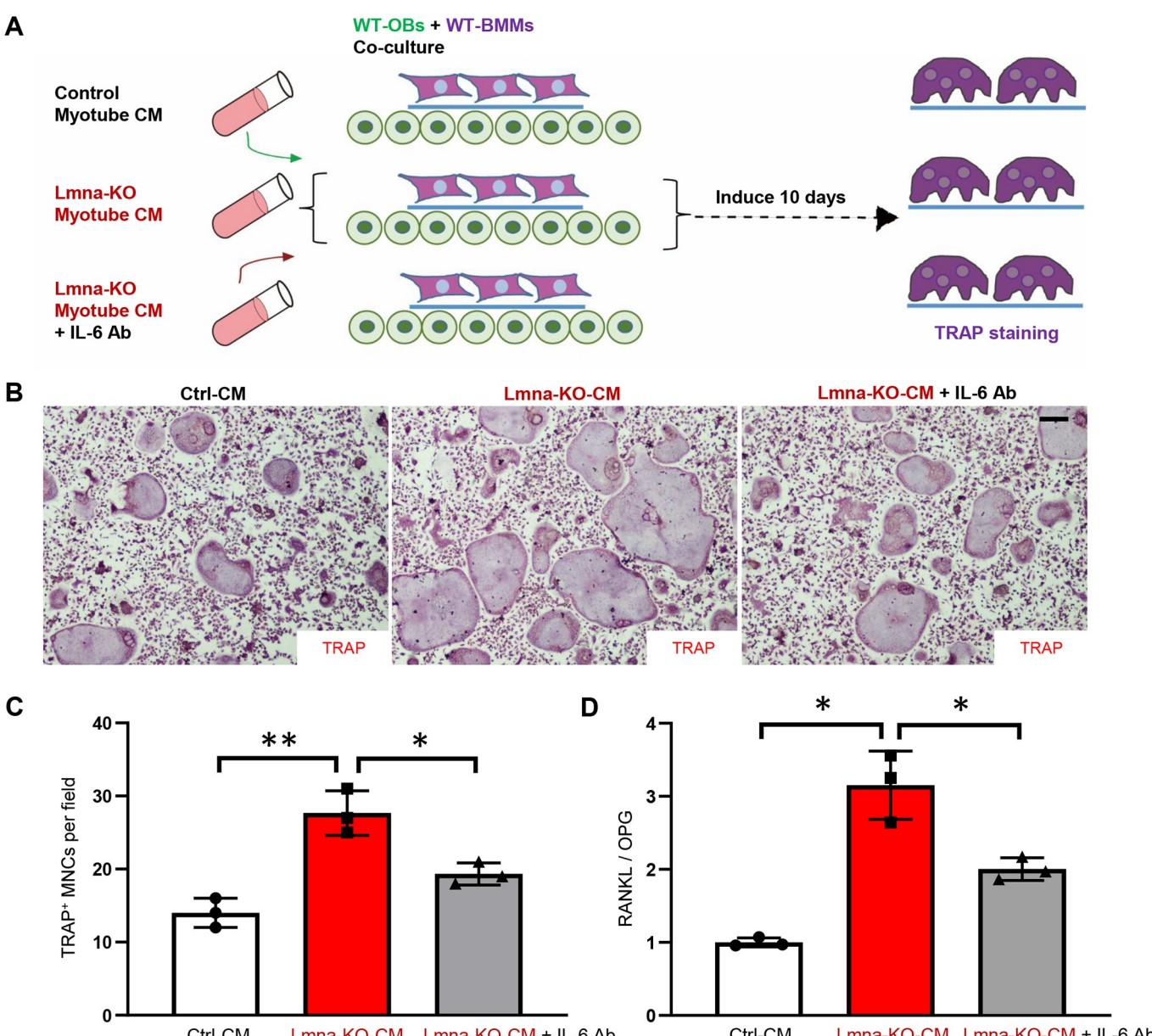

**Fig 6. Diminished osteoclastogenesis in OBs/OCs coculture system in the presence of IL-6 blocking Ab.** (A) Experimental strategy. WT OBs and WT BMMs were treated with CM of myotubes from Ctrl, Lmna-KO, or Lmna-KO plus IL-6 Ab for 10 days. (B) Representative images of the cultures treated with CM for 10 days. Scale bar, 100 μm. (C) The quantitative analyses of TRAP$^+$ MNCs per field. The values shown are means ± SD from three different cultures. $^*P < 0.05$, $^{**}P < 0.01$, significant difference. (D) Real-time PCR analysis of RANKL/OPG expression in OBs treated with CM of Ctrl, Lmna-KO, or Lmna-KO plus IL-6 Ab myotube CM. $^*P < 0.05$, significant difference. The underlying data for this figure can be found in S1 Data. Ab, antibody; BMM, bone marrow macrophage/monocyte; CM, conditioned medium; Ctrl, control; IL, interleukin; KO, knockout; Lmna, lamin A/C gene; NF-κB, nuclear factor κB; MNC, multinucleated cell; OB, osteoblast; OC, osteoclast; OPG, osteoprotegerin; RANKL, receptor activator of NF-κB ligand; TRAP, tartrate-resistant acid phosphatase; WT, wild-type.

This view is further supported by the observations of deficits in twitch and tetanic forces in the IL-6–KO; Lmna$^{HSA}$-cko double mutant mice similar to those of Lmna$^{HSA}$-cko mice (S9D–S9F Fig).

However, the trabecular bone loss in Lmna$^{HSA}$-cko mice (at 3 mo) appeared to be attenuated in the double mutant mice. As shown in Fig 7, μCT analysis of femurs from the 3-mo IL-6–KO; Lmna$^{HSA}$-cko double mutant and Lmna$^{HSA}$-cko single mutant mice showed increases

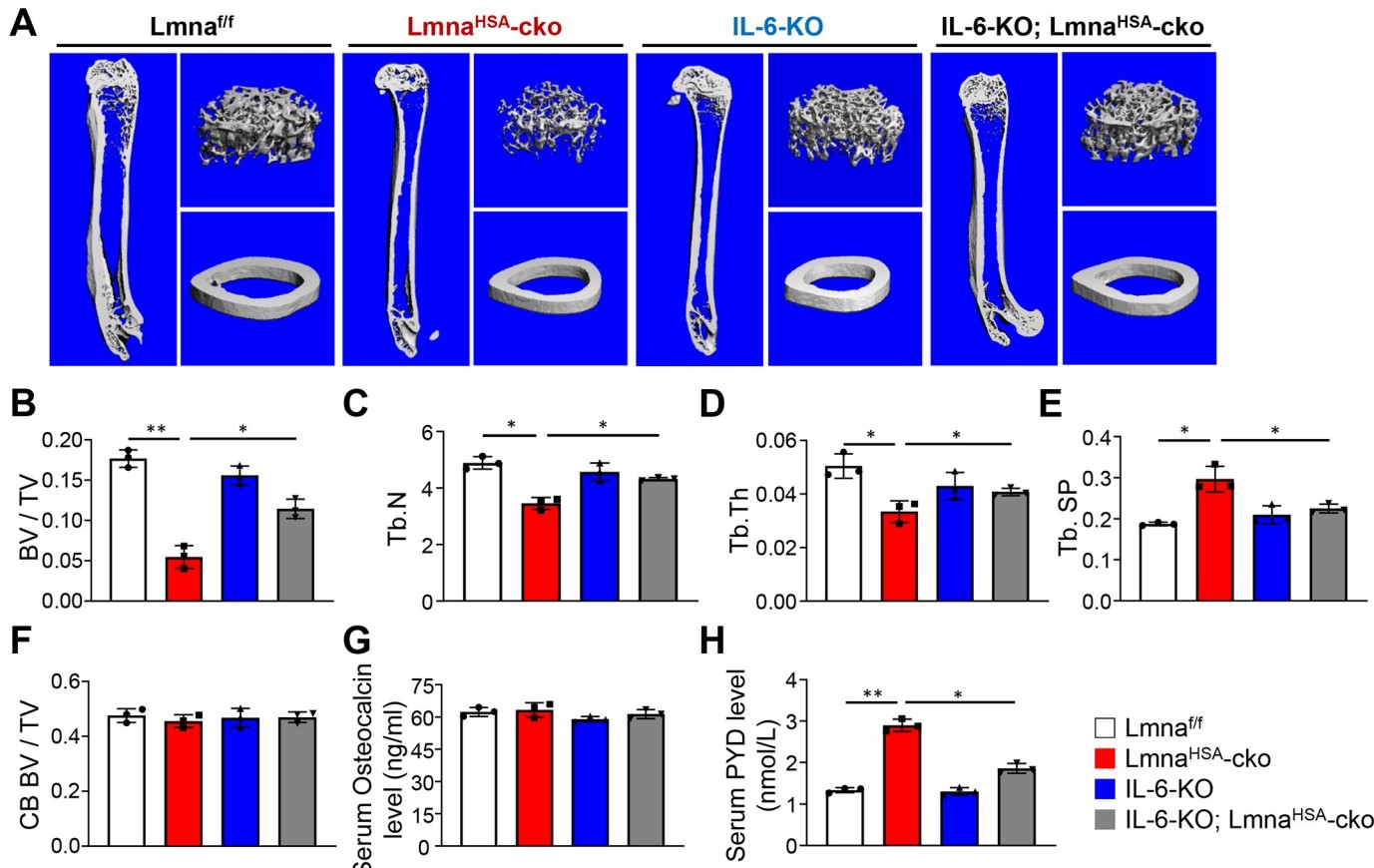

**Fig 7. Attenuated trabecular bone loss phenotype in Lmna^HSA-cko mice by IL-6 KO.** (A) Representative μCT 3D images of femurs from 3-month-old Lmna^f/f, Lmna^HSA-cko, IL-6–KO, and IL-6–KO; Lmna^HSA-cko littermates. (B-F) Quantification analyses by direct model of μCT analysis. Three different male mice of each genotype per group were examined blindly. $^*P < 0.05$, $^{**}P < 0.01$, significant difference. (G) The serum osteocalcin levels were measured by ELISAs. The values shown are means ± SD from three males per genotype. (H) ELISA analysis of serum PYD levels. The values presented are means ± SD ($n = 3$). $^*P < 0.05$, $^{**}P < 0.01$, significant difference. The underlying data for this figure can be found in S1 Data. μCT, microcomputer tomographic; BV/TV, bone volume over total volume; CB, cortical bone; cko, conditional knockout; HSA, human alpha-skeletal actin; IL, interleukin; KO, knockout; Lmna, lamin A/C gene; Lmna^f/f, floxed Lmna mice; Lmna^HSA-cko, skeletal muscle–specific Lmna-cko mice; PYD, pyridinoline; Tb.N, trabecular bone number; Tb.Sp, trabecular bone space; Tb.Th, trabecular bone thickness.

in trabecular BV/TV, Tb.N, and trabecular bone thickness and a decrease in Tb.Sp in the double mutant mice as compared with Lmna^HSA-cko mice (Fig 7A–7F). Notice that the serum Ocn levels in the double mutant mice remained unchanged (Fig 7G) as compared with those in Lmna^HSA-cko mice, but the serum PYD levels were lower in the double mutant mice than those of Lmna^HSA-cko mice (Fig 7H), in line with the view for IL-6's function in promoting bone resorption. In aggregate, these results suggest that the increased osteoclastogenesis in Lmna^HSA-cko mice is likely due to the increased IL-6 expression and secretion by Lmna-KO muscles.

## Exacerbated senescence in Lmna^HSA-cko muscles that is responsible for the increase of IL-6

To understand how muscle lamin A/C regulates IL-6 expression, we wondered whether IL-6 is one of the senescence-associated secretory phenotype (SASP) factors induced by loss of lamin A/C in muscles. This view was in light of the literature report that IL-6 is a key factor of SASP [34] and our observations that several cytokines and growth factors increased in Lmna-KO

myotubes (Fig 5A–5C) appeared to be SASP-like factors. To further test this view, we carried out the following experiments to address whether loss of lamin A/C in muscle cells (C2C12) or muscles in vivo results in an increase of cellular senescence. To this end, we first examined the senescence-associated beta-galactosidase (SA-β-gal) activity, a key marker of cellular senescence, in Lmna-KO muscles and muscle cells by X-gal staining. A marked increase of SA-β-gal activity was detected not only in Lmna-KO muscle cells (S10A and S10B Fig) but also in Lmna-KO skeletal muscles at 3 mo (Fig 8A and 8B). Second, we examined the expression of p16$^{INK4a}$, another key marker of cellular senescence [35], in Lmna-KO muscles and C2C12 muscle cells. The p16$^{INK4a}$ mRNA and protein levels were elevated in Lmna-KO muscles and C2C12 muscle cells by real-time PCR and western blot analyses, respectively (Fig 8C–8E) (S10F–S10H Fig). An increase of p16$^{INK4a}$ in the nucleus of Lmna-KO C2C12 cells was also detected by immunostaining analysis (S10C–S10E Fig). Third, we examined additional senescence-associated markers, including p53, Lamin B1, Bub1-related kinase (BubR1), and NF-kB signaling [36–39], in Lmna-KO C2C12 cells and muscles by western blot analysis. Increases in p53, phospho-NF-kB65, and NF-kB65 and decreases in inhibitor of nuclear factor kappa B (IκBα), Lamin B1, and BubR1 were all detected in Lmna-KO muscles and muscle cells (Fig 8C and 8D) (S10F and S10G Fig). Together, all of these results support the view for an increase of cellular senescence in Lmna$^{HSA}$-cko muscles as well as Lmna-KO muscle cells.

We next addressed whether p16$^{INK4a}$, a driver of cellular senescence [35,36], plays a role in the up-regulation of IL-6 expression in Lmna-KO myotubes. To this end, the p16$^{INK4a}$-knockdown (KD) C2C12 cell line was generated by the use of p16$^{INK4a}$ short hairpin RNA (shRNA) lentivirus. Western blot analysis confirmed the decrease of p16$^{INK4a}$ expression in both control and Lmna-KO muscle cells as compared with that of scramble RNA controls (Fig 8F). ELISA analysis of IL-6 in the CMs of these cells showed a decrease in Lmna-KO; p16$^{INK4a}$–KD muscle cells as compared with that of Lmna-KO cells (Fig 8G). Taken together, these results suggest that the increased p16$^{INK4a}$ or cellular senescence in Lmna-KO muscles is likely to be responsible for the elevated IL-6 expression.

## Discussion

Lamin A/C, a nuclear lamina protein, is implicated in the pathogenesis of various laminopathies. The underlying cellular and molecular mechanisms for laminopathies are complex and remain to be explored. Here, we provide evidence for a crucial role of skeletal muscle lamin A/C in preventing laminopathy-like deficits, which include lower body weight, loss of body fat and muscle mass, and bone loss. Loss of Lmna in skeletal muscles results in not only muscle deficits (losses of muscle mass and strength) but also trabecular bone loss. Further cellular studies reveal that trabecular bone loss is largely due to elevated osteoclastogenesis and osteoclastic bone resorption. Further molecular studies suggest a working model depicted in Fig 8H in which Lmna deficiency in skeletal muscles increases cellular senescence and IL-6 expression and secretion, thus promoting osteoclastogenesis and osteoporotic deficit.

Mutations in *LMNA* gene are identified in a wide variety of disorders, so-called laminopathies, which include HGPS, RD, MADA, EDMD2, atypical progeria syndrome, and Malouf syndrome [9]. A feature of laminopathies is the accelerated aging of various tissues, including adipose, peripheral nerves, muscles, skins, and bone [8]. Partial or generalized loss of subcutaneous fat, muscle weakness, skin abnormalities, osteoporosis, and osteolysis are often observed in patients with laminopathy [25,40]. Although extensive efforts have been invested to understand lamin A/C's function, exactly how lamin A/C prevents osteoporotic deficit remains to be explored. Notice that gene-targeted mice with HGPS mutation (Lmna$^{HG/HG}$ mice, a gene-targeted allele yielding exclusively progerin, a mutant form of pre–lamin A) display laminopathic

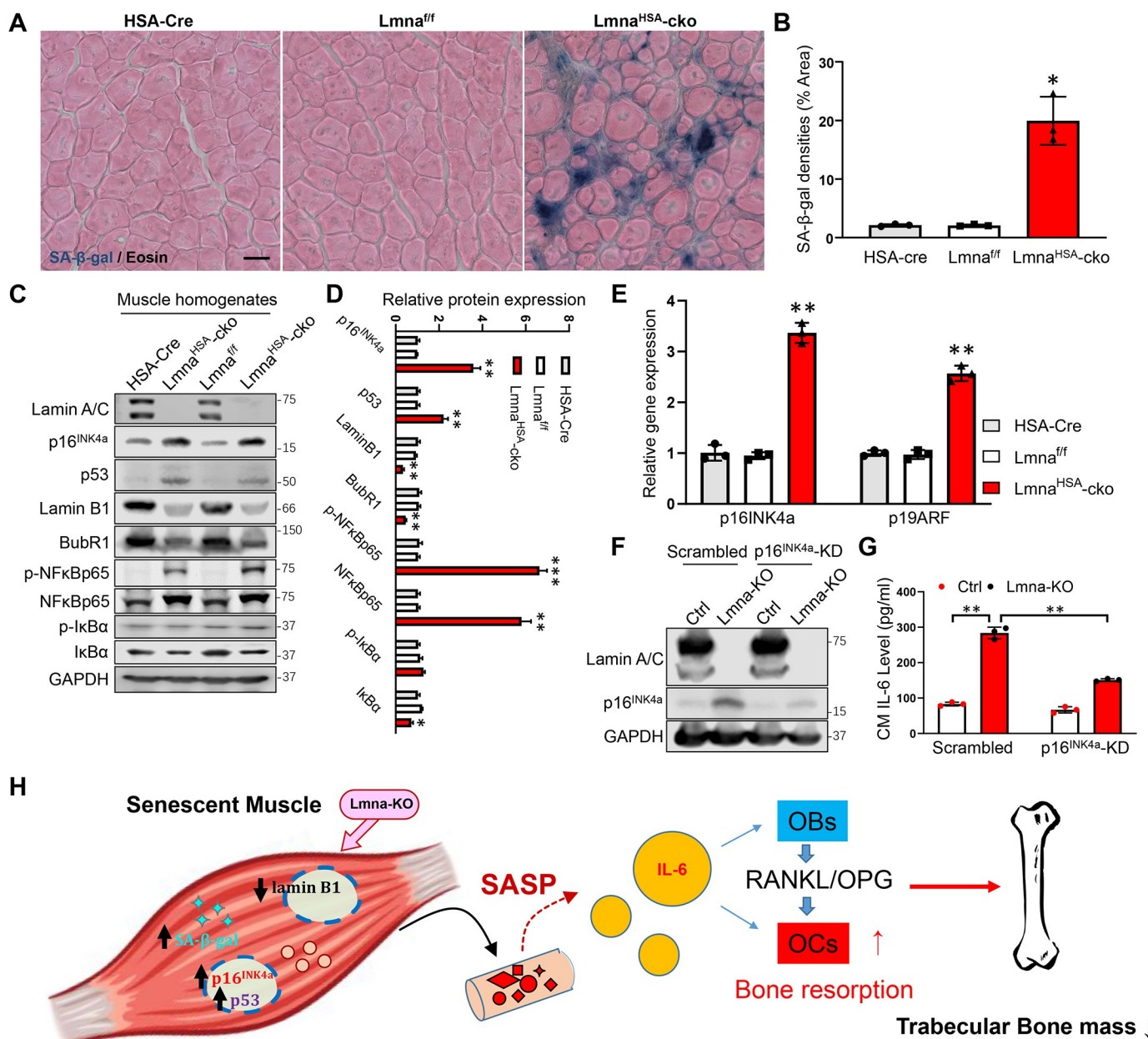

**Fig 8. Increased cellular senescence in Lmna-KO muscles and the requirement of p16$^{INK4a}$ for the increase of IL-6 expression in Lmna-KO muscle cells.** (A) SA-β-gal staining counterstained with eosin of gastrocnemius cross-section from 3-mo HSA-Cre, Lmna$^{f/f}$, and Lmna$^{HSA}$-cko mice. Scale bar, 20 μm. (B) Quantification of SA-β-gal densities (mean ± SD; $n$ = 3). $^*P < 0.05$, significant difference. (C) Western blot analysis of indicated protein expression in muscles from mice with indicated genotypes (at 1 mo). GAPDH was used as a loading Ctrl. (D) Quantification analysis (mean ± SD; $n$ = 3). $^*P < 0.05$, $^{**}P < 0.01$, $^{***}P < 0.001$. (E) Real-time PCR analysis of p16$^{INK4a}$ and p19ARF expression in muscles from mice with indicated genotypes (at 1 mo) (mean ± SD; $n$ = 3). $^{**}P < 0.01$, significant difference. (F) Western blot analysis of p16$^{INK4a}$, which was reduced in p16$^{INK4a}$-KD muscle cells. (G) ELISA analysis of IL-6 level in CM of Ctrl, Lmna-KO, p16$^{INK4a}$-KD, and Lmna-KO; p16$^{INK4a}$-KD C2C12 muscle cells (mean ± SD; $n$ = 3). $^{**}P < 0.01$, significant difference. (H) Illustration of a working model. Lmna loss in skeletal muscles results in increased cellular senescence and p16$^{INK4a}$, which increases the expression and secretion of SASP factors, including IL-6, thus up-regulating RANKL expression in OBs and promoting osteoclastogenesis and trabecular bone loss. The underlying data for this figure can be found in S1 Data. BubR1, Bub1-related kinase; cko, conditional knockout; CM, conditioned medium; Cre, cyclization recombination enzyme; Ctrl, control; GAPDH, glyceraldehyde-3-phosphate dehydrogenase; HSA, human alpha-skeletal actin; IκBα, inhibitor of nuclear factor kappa B; IL, interleukin; KD, knockdown; KO, knockout; Lmna, lamin A/C gene; Lmna$^{f/f}$, floxed Lmna mice; Lmna$^{HSA}$-cko, skeletal muscle–specific Lmna-cko mice; mo, months old; NF-κB, nuclear factor kappa-light-chain-enhancer of activated B cells; OB, osteoblast; OC, osteoclast; OPG, osteoprotegerin; p19ARF, the alternate reading frame tumor-suppressor protein; RANKL, receptor activator of NF-κB ligand; SA-β-gal, senescence-associated beta-galactosidase; SASP, senescence-associated secretory phenotype.

phenotypes as well as osteoporosis and osteolysis [41]. Because the accumulation of pre–lamin A is detected in $Lmna^{HG/HG}$ cells, it is believed that the pre–lamin A is toxic, which may increase the rate of osteoclastogenesis and osteolytic activity over its effect on OB differentiation [42]. Although these studies focus on the effects of pre–lamin A on bone remodeling, it remains unclear or controversial whether the loss of lamin A/C contributes to the osteoporotic deficit [43,44]. Here, we provide evidence for muscle lamin A/C loss to be essential for the osteoporotic deficit and laminopathy-like phenotypes (Fig 1 and Fig 2).

Muscles and bones are anatomically and functionally connected. Besides biomechanical function, skeletal muscles and bones are endocrine organs that can secrete factors modulating biological functions within local or distal environments. For example, skeletal muscles secrete cytokines and growth factors that act as paracrine or endocrine factors on a variety of tissues, including bone [45,46]. Aging-associated osteoporosis frequently coexists with sarcopenia or cachexia [47,48]. Sarcopenia is normally defined as age-associated decreases in muscle mass and function, whereas cachexia is defined as inflammation-mediated loss of muscles and fat, but both of them are wasting disorders. It is unclear how osteoporosis is linked with sarcopenia or cachexia. Our study demonstrates that muscle atrophy induced by Lmna deficiency is sufficient to cause an osteoporosis-like deficit, which is likely due to muscle senescence–driven IL-6 expression.

IL-6 is now perceived as a highly pleiotropic factor influencing many biological events in several organs, especially the immune and the central nervous systems, as well as bone. IL-6 plays a crucial role in bone remodeling, including both OB and OC differentiation and activities [49]. However, the underlying mechanisms remain complex and controversial. Notice that OBs express low levels of IL-6 receptor (IL-6R), and the presence of soluble IL-6R (sIL-6R) is required for the maximum effect of IL-6 on these cells [50–53]. RANKL has been identified as the major cytokine implicated in OB-to-OC communications. RANKL is expressed on stromal cells/OBs, and its interaction with the receptor activator of NF-κB (RANK), present on OC precursors and mature OCs, induces OC differentiation and activation [29]. Our results suggest that IL-6 stimulates OC differentiation and bone resorption by an OB-dependent mechanism. In cocultures containing both OB and OC precursors, IL-6 is likely to induce osteoblastic production of various factors, including RANKL, IL-1, parathyroid hormone–related peptide (PTHrP), and prostaglandin E2 [52,54,55], that promote OC differentiation or activation. However, it is noteworthy that IL-6 is also found to reduce RANKL-induced OC formation and bone resorption obtained from purified OC precursors [56,57]. In our studies, the CM of Lmna-KO myotubes increased the ratio of RANKL/OPG in OBs and thus OC differentiation in OB-BMM cocultures, but it had little to no effect on OC differentiation by its direct treatment on BMMs (Fig 4). In addition, IL-6 blocking antibody diminished Lmna-KO CM–induced osteoclastogenesis in the coculture assays (Fig 6), and IL-6 KO attenuated trabecular bone loss deficit in $Lmna^{HSA}$-cko mice (Fig 7). These results support the view for IL-6 to promote OC differentiation in an OB-dependent fashion. Note that there are many other inflammatory factors or growth factors up-regulated in Lmna-KO muscle (Fig 5C) in addition to IL-6. It is possible that these factors also play a role in OC differentiation [58–61], although we believe that IL-6 is a key factor in linking muscle senescence with osteoclastogenesis and trabecular bone loss. It is of interest to note that patients affected by Duchenne muscular dystrophy (DMD) showed low bone mineral density (BMD) Z-scores and high bone resorption marker and serum IL-6 [62]. In addition, X chromosome–linked muscular dystrophy ($Dmd^{mdx}$) mice also exhibit reduced bone mass and an increased serum level of IL-6 [62–65], similar phenotypes as those in $Lmna^{HSA}$-cko mice. These results suggest that increased IL-6 expression in muscles may be an important factor causing osteoporosis.

In our study, 3-mo OB-selective Lmna-KO mice showed no change in bone mass (S2 Fig). However, the IL-6 mRNA and protein levels in the CM were increased in Lmna-KO BMSCs as compared with those of control BMSCs (S7D and S7E Fig). But, unlike Lmna[HSA]-cko mice (S7A Fig), the serum level of IL-6 showed little to no difference in Lmna[Ocn]-cko mice compared with control mice (S7B Fig). The fold of IL-6 increase in Lmna-KO BMSCs (approximately 1.7-fold in CM) was less than that in Lmna-KO muscle cells (approximately 4-fold) (S7C and S7D Fig). These results suggest that, whereas lamin A/C in OB-lineage cells, as with that in muscle cells, is necessary to suppress IL-6 expression, its impact on IL-6 expression in OB-lineage cells appeared to be less than that in muscle cells. Thus, the increased IL-6 levels in OB-lineage cells may be insufficient to cause bone loss in 3-mo Lmna[Ocn]-cko mice. However, it is possible that the bone loss deficit becomes evident in aged Lmna[Ocn]-cko mice, which we will test in the future.

Cellular senescence has received considerable attention recently as a potentially drug-able target to prevent or treat multiple aging comorbidities, including osteoporosis [66]. Cellular stresses activate the p53/p21 and p16[Ink4a]/retinoblastoma (RB) tumor suppressor pathways to initiate senescence [39]. Downstream of these signals, the senescence response is amplified by a number of mediators, including NF-κb, IL-1β, IL-6, and CCAAT/enhancer-binding protein-β. Senescence cells develop revved-up machinery to secrete proinflammatory cytokines, chemokines, and proteases, termed the SASP, which has deleterious paracrine and systemic effects [39]. Because of the remarkable ability of the SASP to cause "sterile inflammation," even a relatively low abundance of senescent cells is sufficient to cause tissue dysfunction [67]. Targeting senescent cells or inhibiting the production of the proinflammatory SASP of senescent cells was reported to markedly improved bone mass and microarchitecture in trabecular and cortical bone in old mice [35,68,69]. Our study showed that Lmna KO increased cellular senescence in muscle and OB-lineage cells, which enhanced the expression and secretion of many cytokines, including IL-6, and that inhibiting p16[INK4a] could reduce the expression and secretion of IL-6 in Lmna-KO muscle cells, implying that the p16[INK4a]-driven cellular senescence may play an important role in regulating muscle cytokine secretion and bone remodeling. However, although p16[INK4a] is a driver for cellular senescence, it can be activated by other factors, such as reactive oxygen species (ROS) and DNA damage, which also occur in lamin-deficient or lamin-mutant cells [70–73]. Thus, the view that senescence causes IL-6 increase and bone loss in vivo requires further evidence; we hope to further test this view in future experiments.

In aggregate, our results suggest a critical role of skeletal muscle lamin A/C in bone remodeling, revealing a link between muscle dystrophy and osteoporosis in patients with laminopathies.

## Materials and methods

### Ethics statement

All experimental procedures were approved by the Institutional Animal Care and Use Committee at Case Western Reserve University (IACUC, 2017–0121 and 2017–0115), according to the United States National Institutes of Health guidelines.

### Animals and reagents

The Lmna[f/f] mice, kindly provided by Dr. Yixian Zhen (Department of Embryology, Carnegie Science), were crossed with HSA-Cre (from the Jackson Laboratory, catalog #006149) or Ocn-Cre (kindly provided by Dr. Tom Clemens, Johns Hopkins Medical School) transgenic mice to generate muscle- or OB-selective cko mutant mice, Lmna[HSA]-cko or Lmna[Ocn]-cko, respectively. The mutant mice were in the C57BL/6J mouse background (for more than six generations). The IL-6–KO mice, which are in the C57BL/6J background, were purchased from the

Jackson Laboratory (catalog #002650). The IL-6–KO homozygous mice were crossed with Lmna[f/w];HSA-Cre mice to get the heterozygous mice, and the double-KO mice were obtained from the next breeding generation.

Mouse monoclonal antibodies, including lamin A/C (39287, ACTIVE MOTIF), β-actin (A1978, Sigma), GAPDH (NB300-221, Novus), IL-6 (MAB406, R&D), NF-κB p65 (#8242, Cell Signal), p-NFκB p65 (sc-135769, Santa Cruz), p-IκBα (#2859, Cell Signaling), and IκBα (#4814, Cell Signaling), were used. Rabbit polyclonal antibody laminin (L9393, Sigma) and lamin B1 (ab16048, Abcam) were used. Secondary antibodies were purchased from Jackson ImmunoResearch Laboratories, Inc. Calcein (C0875) was obtained from Sigma. Other chemicals and reagents used in this study were of analytical grade.

## In vitro OB/OC-lineage cell cultures

Whole bone marrow cells were flushed from long bones of control and Lmna-deficient mice and plated on 100-mm culture plates in DMEM containing 1% penicillin/streptomycin (P/S) and 10% fetal bovine serum (FBS) for 2 days. For OB-lineage culture, plates with adherent cells were replaced with fresh culture medium every 3 days. After 7 days, the adherent cells (BMSCs) were digested with trypsin, and $1 \times 10^4$ cells per square centimeter were plated for experiments. For OB differentiation, 10 mM β-glycerophosphate and 50 μM L-ascorbic acid-2-phosphate were added to the culture medium.

For OC-lineage culture, nonadherent cells were harvested and subjected to Ficoll-Hypaque gradient centrifugation for purification of BMMs. Cells were plated on 100-mm culture dishes in α-MEM containing 10% FBS, 1% P/S, and 10 ng/ml recombinant M-CSF. For osteoclastogenesis, $5 \times 10^4$ BMMs were incubated with OC differentiation medium containing 10 ng/ml recombinant M-CSF and 100 ng/ml recombinant RANKL. Mature OCs began to form on day 4 and 5 after RANKL treatment. The cells were then subjected to TRAP staining to confirm their OC identity.

For treatment with CM of myotubes, BMSCs or BMMs derived from WT mice were placed onto presterilized glass coverslips in 12-well plates. CMs from control or Lmna-KO myotubes plus 10 mM β-glycerophosphate and 50 μM L-ascorbic acid-2-phosphate were added to 12-well plates containing BMSCs every 2 days. After 14 days, cells were subjected to ALP staining. CMs plus 10 ng/ml recombinant M-CSF and 100 ng/ml recombinant RANKL were added to 12-well plates containing BMMs every 2 days. After 7 days, cells were subjected to TRAP staining.

For OB-BMM coculture, primary OB cultures were prepared from long bones of WT mice. Briefly, small bone pieces were incubated in collagenase solution to remove all remaining soft tissue and adhering cells and then transferred to 60-mm culture dishes containing DMEM medium supplemented with 10% FBS, 1% P/S, 10 mM β-glycerophosphate, and 50 μM L-ascorbic acid-2-phosphate. Culture medium was replaced three times per week. Bone cells will start to migrate from the bone chips after 3–5 days. After 2 weeks, the monolayer is trypsinized by incubating the cells with trypsin solution. OBs were plated on 100-mm tissue culture plates in α-MEM containing 10% FBS and 1% P/S. BMMs derived from WT mice were placed onto presterilized glass coverslips in the 12-well plates. The coverslips were then transferred to the 100-mm plates containing OBs. CMs of myotube cultures of control, Lmna-KO, or Lmna-KO plus IL-6 antibody groups were added to the culture plates. After 10 days, cells on the coverslips were subjected to TRAP staining.

## C2C12 culture and KO mutagenesis

Mouse myoblast C2C12 cells (ATCC Cat#CRL-1772, RRID: CVCL_0188) were cultured in the CM with 20% FBS, 1% chicken embryo extract (100-163P, GEMINI), and 1% P/S in DMEM.

Upon 80% confluence, the cells were rinsed with PBS, and the differentiation medium (4% horse serum, 1% P/S in DMEM) was added for 4–5 days until myotube formation was completed.

To generate Lmna-KO C2C12 cells, the CRISPR-Cas9 plasmid pSpCas9(BB)-2A-GFP was purchased from addgene (#48138). The guidance RNA gATCCCCCAGCTCGGCCTCGT was inserted into the BbsI site and verified by sequencing. The plasmid was transfected into C2C12 cells by lipofectamine 3000 (L3000-015, Invitrogen) at 60%–70% confluence. GFP-positive cells were sorted by FACS and placed into a 96-well plate, with one cell per well. The cells were transferred to six-well plates after 5 days, and their genotypes were examined by genomic DNA sequencing. The KO effect was further verified by western blot, and two of the KO cell lines that can differentiate into myotubes were used for the indicated experiments.

The p16$^{INK4a}$-KD cell line was obtained by infection of C2C12 cells with lentiviral particles encoding scramble control or shRNA-p16$^{INK4a}$ (sc-43988-V, Santa Cruz), respectively. In brief, cells were infected with the lentivirus particles for 1 day in 2 μg/ml polybrene medium. At day 3, the culture medium was removed and replaced with complete medium (without polybrene). After 5–6 days, stable clone expressing the shRNA was selected via 5 μg/ml puromycin dihydrochloride, which induces the death of untransduced cells.

## Cell/tissue lysis and western blot

Cells or tissue was lysed in lysis buffer containing 50 mM Tris-HCl (pH 7.5), 150 mM NaCl, 1% (v/v) Triton X-100, 0.1% SDS, 0.5% deoxycholate, and 1 mM EDTA, supplemented with protease inhibitors (1 μg/mL leupeptin and pepstatin, 2 μg/mL aprotinin, and 1 mM PMSF) and phosphatase inhibitors (10 mM NaF and 1 mM Na3VO4). Whole-cell/tissue extracts were fractionated by SDS-PAGE and transferred to a nitrocellulose membrane (Bio-Rad). After incubation with 5% milk in TBST (10 mM Tris, 150 mM NaCl, 0.5% Tween 20 [pH 8.0]) for 1 hour, the membrane was incubated with indicated antibodies overnight at 4°C. Membranes were washed with TBST three times and incubated with a 1:5,000 dilution of horseradish peroxidase–conjugated anti-mouse or anti-rabbit antibodies for 1 hour. Blots were washed with TBST three times and developed with the ECL system (Bio-Rad).

## Immunofluorescence staining and imaging analysis

For muscle cross-section staining, muscles were fixed with 4% PFA in PBS at 4°C overnight. After dehydration by 30% sucrose at 4°C overnight, muscles were frozen at −80°C in the cryo-embedding medium (Ted Pella) and cut into 30-μm sections on a cryostat (HM550, Thermo Scientific). For cell immunofluorescence staining, cells on coverslips were fixed with 4% PFA at room temperature for 20 minutes. Muscle sections or cells were permeabilized with 0.15% Triton X-100 for 8 minutes and then subjected to coimmunostaining analysis using the indicated antibodies. Stained sections or cells were washed three times with PBS and mounted with VECTASHIELD (H-1500; Vector Laboratories) and imaged by confocal microscope at room temperature. Fluorescent quantification was performed using ZEN software according to the manufacturer's instructions (Carl Zeiss).

## In vivo twitch and tetanic force measurement

Torque muscle strength was measured on male mice as previously reported [74]. Briefly, mice were anesthetized with isoflurane continuously supplied by the VetFlo anesthesia system (Kent Scientific) and placed on a 37°C heating pad. The knees were fixed by gently pressing the knee clamps, and the feet were fixed onto the footplate that was connected to the servomotor (Aurora Scientific 1300A). The angle of the footplate was usually at 17° but could be

adjusted for maximal twitch force. The sciatic nerve was exposed at the thigh level and stimulated at 5 mA by two needle electrodes. The stimulation pulse width was 0.2 milliseconds for all experiments. Tetanic contractions were induced by 300-millisecond stimuli at frequencies of 50, 100, 150, and 200 Hz. Each tetanic contraction was followed by an interval of ≥2 minutes. Twitch and tetanic forces were normalized by body weight.

## µCT

The µCT analyses were carried out as described previously [27]. Excised femurs from mice were scanned using the Scanco µCT40 desktop cone-beam µCT scanner (Scanco Medical AG, Brüttisellen, Switzerland, using µCT Tomography v.5.44). Scans were automatically reconstructed into 2D slices, and all slices were analyzed using the µCT Evaluation Program (v.6.5–2, Scanco Medical). The femur was placed inverted in a 12-mm-diameter scanning holder and scanned at the following settings: 12-µm resolution, 55 kVp, and 145 µA with an integration time of 200 milliseconds. For the cortical analysis, the bone was scanned at the midshaft of the bone for a scan of 25 slices. The region of interest (ROI) was drawn on every slice and fitted to the outside of the cortical bone to include all of the bone and marrow. The threshold for cortical bone was set at 329. The 3D reconstruction (µCT Ray v.3.8) was performed using all the outlined slices. Data were obtained on bone volume, total volume, BV/TV, bone density, and cortical thickness. For the trabecular bone, the scan was started at the growth plate and consisted of 211 slices. The ROI was outlined starting below the growth plate (for the femurs from 1-mo mice) and where the condyles ended (for the femurs from older mice). One-hundred slices were outlined from this point, on the inside of the cortical bone, enclosing only the trabecular bone and marrow. Trabecular bone was thresholded at 245, and the 3D analysis was performed on the 100 slices. Data were obtained on bone volume, density, total volume, trabecular number, thickness, and separation.

## Bone histomorphometric analysis

Bone histomorphometric analyses were carried out as previously described [75]. In brief, mouse tibia and femurs were fixed overnight in 10% formalin, decalcified in 14% EDTA, embedded in paraffin, sectioned, and subjected to HE and TRAP staining. Morphometric perimeters were determined by measuring the areas situated at least 0.5 mm from the growth plate, excluding the primary spongiosa and trabeculae connected to the cortical bone.

## Measurements of serum levels of Ocn, PYD, calcium, and IL-6

Blood samples were collected, allowed to clot for 30 minutes, and centrifuged for 10 minutes at 3,000 rpm. Serum was frozen at −80°C until use. Ocn, PYD, and calcium were measured in duplicate with Ocn Elisa kit (Biomedical Technologies, Inc.), MicroVue Serum PYD Elisa kit (Quidel Corporation), and Calcium Detection Kit (Abcam), respectively, as described previously [76]. Serum IL-6 was measured by Mouse IL-6 Quantikine ELISA Kit (R&D Systems) according to the manufacturer's instructions. Concentrations were obtained by comparing readings against standard curves.

## Mouse Cytokine Array

Cell cultured medium was collected and centrifuged for 10 minutes at 1,000 rpm to remove particulates. Medium was assayed immediately or divided into aliquots and stored at −80°C. Cytokines were measured in duplicate with Mouse Cytokine Array Panel A (ARY006, R&D Systems). Briefly, cell culture medium is mixed with a cocktail of biotinylated detection

antibodies. The sample/antibody mixture is then incubated with the Mouse Cytokine Array membrane. Any cytokine/detection antibody complex present is bound by its cognate immobilized capture antibody on the membrane. Following a wash to remove unbound material, streptavidin–horseradish peroxidase and chemiluminescent detection reagents are added sequentially. Light is produced at each spot in proportion to the amount of cytokine bound.

### SA-β-gal staining of muscle and C2C12 muscle cells

Muscle SA-β-gal staining was performed as previously reported [77]. In brief, the gastrocnemius muscle was directly cryosectioned in 30-μm sections. The fresh samples were fixed in PBS containing 1% PFA and 0.2% glutaraldehyde for 10 minutes and then washed with PBS twice and incubated in PBS (pH 5.5) for 30 minutes. The sections were incubated in the X-gal solution containing 4 mM $K_3Fe(CN)_6$, 4 mM $K_4Fe(CN)_6$, 2 mM $MgCL_2$, and 400 μg/ml X-gal in PBS (pH 5.5) in the dark at 37˚C for 24 hours. Slides were washed with PBS three times, fixed in 1% PFA for 30 minutes, and then counterstained with 0.2% eosin. Slides were mounted with the aqueous mounting medium before taking the images. For muscle cell staining, the control and Lmna-KO C2C12 cell lines were incubated with differentiation medium containing 1% horse serum for 5 days, and the SA-β-gal staining was performed using a SA-β-gal staining kit (Cell Signaling, #9860) according to the manufacturer's instructions.

### RNA isolation and real-time PCR

Total RNA was isolated by Trizol extraction (Invitrogen, Carlsbad, CA, USA). Q-PCR was performed by using Quantitect SYBR Green PCR Kit (Bio-Rad) with a Real-Time PCR System (Opticon Monitor 3). The following primers were used: RANKL, 5′-ATCCCATCGGGTTCC CATAA-3′ and 5′-TCCGTTGCTTAACGTCATGTTAG-3′; OPG, 5′-GGCCTGATGTATGC CCTCAA-3′ and 5′-GTGCAGGAACCTCATGGTCTTC-3′; IL-1ra, 5′- CCCTGCAAGATG CAAGCC-3′ and 5′- CCGTGGATGCCCAAGAAC-3′; IL-6, 5′- CTTGGGACTGATGCTGG TG-3′ and 5′- TTGGGAGTGGTATCCTCTGTGA-3′; IL-7, 5′-GCCCGAATAATGAACCA A-3′ and 5′-TTGCGAGCAGCACGATTT-3′; IL-15, 5′- GGAATACATCCATCTCGTGCTA CTT-3′ and 5′- TCTTACATCTATCCAGTTGGCCTCT-3′; IL-16, 5′- TGAGTGAGAACCA TAGCCATAC-3′ and 5′- CTGCCATCTGCTGACTTGTA-3′; M-CSF, 5′- GCTGCTGCTGG TCTGTCT-3′ and 5′- GGCAATCTGGCATGAAGT-3′; MIG, 5′- CCTCTTGGGCATCATC TT-3′ and 5′- TGAGGTCTTTGAGGGATTT-3′; TNF-a, 5′- GGCGGTGCCTATGTCTCA-3′ and 5′- CCTCCACTTGGTGGTTTGT-3′; I-309, 5′- GCTTACGGTCTCCAATAGC-3′ and 5′- CAGCCTGAATACCACAGC-3′; RANTES, 5′- ACCACTCCCTGCTGCTTT-3′ and 5′- A CACTTGGCGGTTCCTTC-3′; BMP1, 5′-CGGCGATAACAACTACCC-3′ and 5′-GTCAGT CTCCTCCTCCACC-3′; BMP2, 5′- TGTGAGGATTAGCAGGTCTTTGC-3′ and 5′- CTCGT TTGTGGAGCGGATGT-3′; BMP4, 5′- GAGGAGTTTCCATCACGAAG-3′ and 5′- TCTCC ACTCCCTTGAGGTAA-3′; BMP6, 5′- CCCTGTCCAATGACGACGAAG-3′ and 5′- CCAG GAGACTCTTGCGGTTCA-3′; IGF-1, 5′-TCTACCTGGCGCTCTGCT-3′ and 5′-CCTGTG GGCTTGTTGAAGT-3′; FGF-2, 5′-CGAGAAGAGCGACCCACA-3′ and 5′-AGAAGCCAG CAGCCGTCCA-3′; TGFβ-1, 5′-ACCGCAACAACGCCATCT-3′ and 5′-GGGCACTGCTTC CCGAAT-3′; myostatin, 5′-TCAAACCCATGAAAGACG-3′ and 5′-CTGCCAAATACCAG TGCC-3′; p16INK4A, 5′-TGCGGGCACTGCTGGAAG-3′ and 5′-ACCGGGCGGGAGAAG GTAG-3′; P19ARF, 5′-ATGGGTCGCAGGTTCTTG-3′ and 5′-GTGCGGCCCTCTTCTCA A-3′; GAPDH, 5′-AAGGTCATCCCAGAGCTGAA-3′ and 5′-CTGCTTCACCACCTTCTT GA-3′; β-actin primers 5′-AGGTCATCACTATTGGCAACGA-3′ and 5′-CATGGATGCCA CAGGATTCC-3′ were used for normalization.

## Statistical analysis

All data were expressed as means ± SD. For in vivo studies, three to six mice per genotype per assay were used. For in vitro cell biological and biochemical studies, each experiment was repeated at least three times. Data were analyzed by Student $t$ test, two-way ANOVA, and post hoc test (GraphPad Prism 8). The significance level was set at $P < 0.05$ (*$P < 0.05$, **$P < 0.01$, ***$P < 0.001$).

## Supporting information

**S1 Fig. Generation of OB- and muscle-selective Lmna-KO mice.** (A) Strategy to cleave exon 2 of *Lmna* flanked by loxP sites in the Lmna$^{flox}$ allele. Lmna$^{flox/flox}$ mice were crossed with Ocn-Cre or the HSA-Cre transgenic mice to generate OB- and muscle-selective cko mutant mice, Lmna$^{Ocn}$-cko and Lmna$^{HSA}$-cko. (B) Western blotting analysis of lamin A/C expression in primary cultured BMSCs and BMMs from 3-mo Lmna$^{f/f}$ and Lmna$^{Ocn}$-cko mice. β-Actin was used as the loading control. (C) Western blotting analysis of lamin A/C expression in different tissue of 3-mo Lmna$^{f/f}$ and Lmna$^{HSA}$-cko mice. GAPDH was used as the loading control. (D) Western blotting analysis of lamin A/C expression in primary cultured BMSCs and BMMs from 3-mo Lmna$^{f/f}$ and Lmna$^{HSA}$-cko mice. β-Actin was used as the loading control. BMM, bone marrow macrophage/monocyte; BMSC, bone marrow stromal cell; cko, conditional knockout; Cre, cyclization recombination enzyme; HSA, human alpha-skeletal actin; KO, knockout; Lmna, lamin A/C gene; Lmna$^{f/f}$, floxed Lmna mice; Lmna$^{HSA}$-cko, skeletal muscle–specific Lmna-cko mice; Lmna$^{Ocn}$-cko, OB-selective Lmna–conditional knockout mice; mo, months old; OB, osteoblast; Ocn, osteocalcin.
(TIF)

**S2 Fig. No change of bone mass in 3-mo Lmna$^{Ocn}$-cko mice.** (A) Photo of a 3-mo Lmna$^{Ocn}$-cko mouse and a littermate Lmna$^{f/f}$ mouse. (B) No change of body weight in the Lmna$^{Ocn}$-cko mouse. (C) Representative HE-stained sections of skin from 3-mo Lmna$^{f/f}$ and Lmna$^{Ocn}$-cko mice. Scale bar, 200 μm. (D,E) Quantification analyses of adipocyte size and subcutaneous fat thickness of 3-mo Lmna$^{f/f}$ and Lmna$^{Ocn}$-cko mice. $N = 4$. (F) Thorax of 3-mo Lmna$^{f/f}$ and Lmna$^{Ocn}$-cko mice. (G) The μCT analysis of femurs from 3-mo Lmna$^{f/f}$ and Lmna$^{Ocn}$-cko littermates. Three different male mice of each genotype per group were examined blindly. (H-L) Quantification analyses ($n = 3$) of TB BV/TV, Tb.N, Tb.Th, Tb.Sp, and CB BV/TV by the direct model of μCT analysis. The underlying data for this figure can be found in S1 Data. μCT, microcomputer tomographic; BV/TV, bone volume over total volume; CB, cortical bone; cko, conditional knockout; Lmna, lamin A/C gene; Lmna$^{f/f}$, floxed Lmna mice; Lmna$^{Ocn}$-cko, OB-selective Lmna–conditional knockout mice; mo, months old; OB, osteoblast; Ocn, osteocalcin; TB, trabecular bone; Tb.N, trabecular bone number; Tb.Sp, trabecular bone space; Tb.Th, trabecular bone thickness.
(TIF)

**S3 Fig. Decreased muscle size and bone mass in 3-mo Lmna$^{HSA}$-cko mice.** (A) Representative images of gastrocnemius cross sections from 3-mo HSA-Cre, Lmna$^{f/f}$, and Lmna$^{HSA}$-cko mice. Scale bar, 20 μm. (B,C) Quantification analyses of cross-section area and central nuclei distribution. $N = 3$ mice per group. *$P < 0.05$, **$P < 0.01$. (D) Histomorphological examinations of the femur from 3-mo HSA-Cre, Lmna$^{f/f}$, and Lmna$^{HSA}$-cko mice by HE staining analysis. Scale bar, 300 μm. (E) Quantification analysis of data from (D). $N = 3$ mice, *$P < 0.05$. The underlying data for this figure can be found in S1 Data. cko, conditional knockout; Cre, cyclization recombination enzyme; HSA, human alpha-skeletal actin; Lmna, lamin A/C gene; Lmna$^{f/f}$, floxed Lmna mice; Lmna$^{HSA}$-cko, skeletal muscle–specific Lmna-cko mice; mo,

months old; Ocn, osteocalcin.
(TIF)

**S4 Fig. No change in muscle size and force, but a decrease in TB mass in 1-mo Lmna$^{HSA}$-cko mice.** (A) Representative images of gastrocnemius cross sections. Scale bar, 20 μm. (B,C) Quantification analyses of cross-section area and central nuclei distribution. $N = 3$ mice per group. (D) Representative twitch curves and tetanic curves at stimulation frequencies 50 and 150 Hz by muscle stimulation. (E,F) Quantification analyses of twitch force and tetanic force. $N = 4$ mice per group. (G) The μCT analysis of femurs from 1-mo Lmna$^{f/f}$ and Lmna$^{HSA}$-cko littermates. Four different male mice of each genotype per group were examined blindly. (H) Quantification analyses ($n = 4$) of TB BV/TV, Tb.N, Tb.Th, Tb.Sp, and CB BV/TV by the direct model of μCT analysis. Data is determined by two-way ANOVA. $^{**}P < 0.01$, $^{***}P < 0.001$, significant difference. The underlying data for this figure can be found in S1 Data. μCT, microcomputer tomographic; BV/TV, bone volume over total volume; CB, cortical bone; cko, conditional knockout; HSA, human alpha-skeletal actin; Lmna, lamin A/C gene; Lmna$^{f/f}$, floxed Lmna mice; Lmna$^{HSA}$-cko, skeletal muscle–specific Lmna-cko mice; mo, months old; TB, trabecular bone; Tb.N, trabecular bone number; Tb.Sp, trabecular bone space; Tb.Th, trabecular bone thickness.
(TIF)

**S5 Fig. Generation of Lmna-KO C2C12 myoblasts.** (A) Sequencing data showing the frameshift or/and terminal codon generated by NHEJ in Lmna-KO cell lines 1–1 and 1–3. (B) Western blot showing KO of lamin A/C in C2C12 cell lines. cko, conditional knockout; KO, knockout; Lmna, lamin A/C gene; NHEJ, nonhomologous end joining.
(TIF)

**S6 Fig. Direct treatment with CM of Lmna-KO myotubes does not affect the differentiation of BMSCs but up-regulates RANKL/OPG expression.** (A) Experimental strategy. (B) ALP staining analysis of cultured BMSCs in the presence of CM of Ctrl or Lmna-KO myotube CM. Scale bar, 100 μm. (C) Quantitative data of the average ALP activities (mean ± SD from five different cultures). (D-F) Real-time PCR analysis of RANKL and OPG expression in OBs treated with CMs of Ctrl or Lmna-KO myotubes. $^{**}P < 0.01$, significant difference from the Ctrl. The underlying data for this figure can be found in S1 Data. ALP, alkaline phosphatase; BMSC, bone marrow stromal cell; CM, conditioned medium; Ctrl, control; KO, knockout; Lmna, lamin A/C gene; OB, osteoblast; OPG, osteoprotegerin; RANKL, receptor activator of nuclear factor κB ligand.
(TIF)

**S7 Fig. Normal serum level of IL-6 in 3-mo Lmna$^{Ocn}$-cko mice.** (A) ELISA analysis of serum IL-6 level in 3-mo Lmna$^{f/f}$ and Lmna$^{HSA}$-cko mice. $^{*}P < 0.05$, significant difference. (B) Serum IL-6 level in 3-mo Lmna$^{f/f}$ and Lmna$^{Ocn}$-cko mice. (C) ELISA analysis of IL-6 level in the culture medium of ctrl and Lmna-KO myotubes induced from C2C12. $^{***}P < 0.001$, significant difference. (D) IL-6 levels in the culture medium of BMSCs derived from 3-mo Lmna$^{f/f}$ and Lmna$^{Ocn}$-cko mice. $^{**}P < 0.01$, significant difference. (E) Real-time PCR analysis of gene expression in BMSCs derived from 3-mo Lmna$^{f/f}$ and Lmna$^{Ocn}$-cko mice. $^{*}P < 0.05$, significant difference. The underlying data for this figure can be found in S1 Data. BMSC, bone marrow stromal cell; cko, conditional knockout; ctrl, control; HSA, human alpha-skeletal actin; IL, interleukin; KO, knockout; Lmna, lamin A/C gene; Lmna$^{f/f}$, floxed Lmna mice; Lmna$^{HSA}$-cko, skeletal muscle–specific Lmna-cko mice; Lmna$^{Ocn}$-cko, OB-selective Lmna–conditional knockout mice; mo, months old; OB, osteoblast; Ocn, osteocalcin.
(TIF)

**S8 Fig. Generation of IL-6–KO and Lmna^HSA-cko mice.** (A) Mouse images of indicated genotypes at 3 mo. The Lmna^HSA-cko and IL-6–KO; Lmna^HSA-cko mice showed kyphotic phenotype. (B) ELISA analysis of serum levels of IL-6 in 3-mo mice. $^*P < 0.05$, $^{**}P < 0.01$, significant difference. The underlying data for this figure can be found in S1 Data. cko, conditional knockout; HSA, human alpha-skeletal actin; IL, interleukin; KO, knockout; Lmna, lamin A/C gene; Lmna^HSA-cko, skeletal muscle–specific Lmna-cko mice; mo, months old. (TIF)

**S9 Fig. Negligible amelioration of muscle loss in Lmna^HSA-cko mice by IL-6 depletion.** (A) Representative images of gastrocnemius cross sections. Scale bar, 20 μm. (B,C) Quantification analyses of cross-section area and central nuclei distribution. $N = 5$ mice per group. $^*P < 0.05$, $^{**}P < 0.01$. (D) Representative twitch curves and tetanic curves at stimulation frequencies 50 and 150 Hz by muscle stimulation. (E,F) Quantification analyses of twitch force and tetanic force. $^{**}P < 0.01$. $N = 4$ mice per group. The underlying data for this figure can be found in S1 Data. cko, conditional knockout; HSA, human alpha-skeletal actin; IL, interleukin; Lmna, lamin A/C gene; Lmna^HSA-cko, skeletal muscle–specific Lmna-cko mice. (TIF)

**S10 Fig. Increased cellular senescence in Lmna-KO myotubes.** (A) Representative images of SA-β-gal staining of ctrl and Lmna-KO muscle cells. Scale bar, 20 μm. (B) Quantification of SA-β-gal$^+$ cells. $^{***}P < 0.001$. (C) Immunostaining analysis of p16^INK4a in ctrl and Lmna-KO C2C12 cells. (D,E) Quantification analysis (means ± SD, $n = 20$ cells from five different assays, $^{***}P < 0.001$). (F) Western blot analysis of indicated protein expression in ctrl and Lmna-KO muscle cells. GAPDH was used as the loading ctrls. (G) Quantification analysis (mean ± SD; $n = 3$). $^*P < 0.05$, $^{**}P < 0.01$. (H) Real-time PCR analysis of p16INK4a and p19ARF expression in ctrl and Lmna-KO muscle cells. $^*P < 0.05$, significant difference. The underlying data for this figure can be found in S1 Data. ctrl, control; KO, knockout; Lmna, lamin A/C gene; p19ARF, the alternate reading frame tumor-suppressor protein; SA-β-gal, senescence-associated beta-galactosidase. (TIF)

**S1 Data Underlying numerical data and statistical analysis for figure panels 1B, E, F, H, I, K, L; 2B, C, E, F, H; 3A, C, D, E, F, G, I, K; 4C, F; 5B, C, E; 6C, D; 7B, C, D, E, F, G, H; 8B, D, E, G; S2B, D, E, H, I, J, K, L; S3B, C, E; S4B, C, E, F, H; S6C, D, E, F; S7A, B, C, D, E; S8B; S9B, C, E, F; and S10B, D, E, G, H.** (XLSX)

**S1 Raw Images. Original images supporting all blot and gel results reported in Figs 5A, 5D; 8C, 8F; S1B, S1C, S1D; S5B and S10F.** The loading order, experimental samples, and molecular weight were indicated. The lanes used in the final figure were marked with a yellow box, and the lanes not used were marked with an "X" above. (PDF)

## Acknowledgments

We thank Dr. Yixian Zhen (Department of Embryology, Carnegie Science) for providing us the Lmna^flox/flox mice, Ms. Xue-Mei Cao and Dr. Maria S. Johnson (University of Alabama at Birmingham) for μCT analysis, Ms. Shan Xiong for carefully revising the grammatical and syntactic errors in the manuscript, and members of the Xiong and Mei laboratories for helpful discussions.

## Author Contributions

**Conceptualization:** Lei Xiong, Kai Zhao, Yu Cao, Lin Mei, Wen-Cheng Xiong.

**Data curation:** Lei Xiong, Kai Zhao, Hao-Han Guo, Wen-Cheng Xiong.

**Formal analysis:** Lei Xiong, Kai Zhao, Wen-Cheng Xiong.

**Funding acquisition:** Lin Mei, Wen-Cheng Xiong.

**Investigation:** Lei Xiong, Kai Zhao, Wen-Cheng Xiong.

**Methodology:** Lei Xiong, Kai Zhao, Yu Cao, Hao-Han Guo, Jin-Xiu Pan, Xiao Yang, Xiao Ren, Wen-Cheng Xiong.

**Project administration:** Lin Mei, Wen-Cheng Xiong.

**Resources:** Lei Xiong, Wen-Cheng Xiong.

**Software:** Lei Xiong, Kai Zhao, Xiao Ren.

**Supervision:** Lei Xiong, Lin Mei, Wen-Cheng Xiong.

**Validation:** Lei Xiong, Kai Zhao, Wen-Cheng Xiong.

**Visualization:** Lei Xiong, Kai Zhao, Wen-Cheng Xiong.

**Writing – original draft:** Lei Xiong.

**Writing – review & editing:** Lei Xiong, Wen-Cheng Xiong.

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
