## [Editor Report · Decision Letter 0]

20 Dec 2019

Dear Dr Xiong, 

Thank you for submitting your manuscript entitled "Linking skeletal muscle aging with osteoporosis by Lamin A/C –deficiency" for consideration as a Research Article by PLOS Biology.

Your manuscript has now been evaluated by the PLOS Biology editorial staff as well as by an academic editor with relevant expertise and I am writing to let you know that we would like to send your submission out for external peer review.

Please re-submit your manuscript within two working days, i.e. by Dec 22 2019 11:59PM.

***Please be aware that, due to the voluntary nature of our reviewers and academic editors, manuscripts may be subject to delays due to their limited availability during the holiday season. Please also note that the journal office will be closed entirely 21st- 29th December inclusive, and 1st January 2020. Thank you for your patience.***

Kind regards,

Di Jiang

PLOS Biology

---

## [Decision Letter · Decision Letter 1]

28 Jan 2020

Dear Dr Xiong,

Thank you very much for submitting your manuscript "Linking skeletal muscle aging with osteoporosis by Lamin A/C –deficiency" for consideration as a Research Article at PLOS Biology. Your manuscript has been evaluated by the PLOS Biology editors, an Academic Editor with relevant expertise, and by three independent reviewers.

In light of the reviews (below), we would welcome re-submission of a much-revised version that will address all the reviewers' comments including, our academic editor stresses, the issue of circulating IL-6 levels which the first 2 reviewers requested, looking at NF-kB activation and p53 activation requested by reviewer 1, the concern from reviewer 3 regarding senescence in muscle as being a cause for increased IL-6 expression, and the point made by reviewer 3 about distinguishing between the loss of Lamin A versus dominant missence mutations that cause most laminopathies. We cannot make any decision about publication until we have seen the revised manuscript and your response to the reviewers' comments. Your revised manuscript is also likely to be sent for further evaluation by the reviewers.

We expect to receive your revised manuscript within 2 months. 

*NOTE: In your point by point response to to the reviewers, please provide the full context of each review. Do not selectively quote paragraphs or sentences to reply to. The entire set of reviewer comments should be present in full and each specific point should be responded to individually, point by point.

Sincerely,

Di Jiang

PLOS Biology

REVIEWS:

Reviewer #1: L. Xiong et al. provide evidence that skeletal muscle expressed Lamin A/C plays essential roles in preventing not only muscle aging, but also bone loss. The bone loss appears to be due in large to the increase in osteoclast mediated bone resorption. They further carried out mechanistic studies, in particular, antibody array screening for factors secreted from muscle cells, and they identified an increase of IL-6, a pro-inflammatory cytokine, in LaminA/C deficient muscle cells. Moreover, they provide evidence that IL-6 expression is necessary for muscle Lamin A/C-deficiency induced bone loss. Finally, they showed that loss of Lamin A/C in muscles results in senescence and senescence associated secretory phenotype (SASP), which contribute to the increase of IL-6. These are novel findings, and the work provide important insights into muscle Lamin A/C's function, muscle senescence, IL-6, and muscle-bone cross talks. The experiments are well-designed and executed. The authors should consider the following concerns:

1, In Fig S2, 3-month old OB-selective Lmna KO mice showed no change in bone mass. Does LmninA/C regulate IL-6 expression in osteoblasts? If so, why their bone masses do not change? It is necessary to examine IL-6 levels in the serum and osteoblasts of LmnaOcn-cko mice.

2. The nuclear factor-κB (NF-κB) signaling is a major pathway that stimulates SASP, and the expression of multiple cytokines. Is it altered in Lmna-KO muscles or muscle cells?

3. In Fig 8B, in addition to muscle samples, it is necessary to examine p53, another critical transcription factor involved in senescence and cell death, in Lmna-KO muscle cells.

Reviewer #2: The manuscript submitted by Lei Xiong et al is an interesting and well designed study aimed at the evaluation of LMNA-linked pathomechanisms causing bone degeneration. The authors take advantage of two different mouse models lacking Lmna expression in skeletal muscle or osteoblasts and show that Lmna-null myoblasts undergo cellular senescence and activated a senescence-associated secretory phenotype featuring high interleukin 6 (IL-6) levels. The authors show that Il-6 secretion determines osteoclastogenesis and bone resorption. Involvement of Il-6 in osteoclastogenesis has been previously proven, thus loss of lamin A/C appears to trigger a known mechanism of bone degeneration, which is selectively inhibited by blocking IL-6. The reported data are extremely interesting from a mechanistic point of view. However, several caveats need to be addressed:

1) loss of LMNA expression is neither the condition of muscular dystrophies nor the condition of progeroid syndromes linked to LMNA. This means that any conclusion should be validated in the presence of a pathogenetic LMNA mutation, a condition more close to human laminopathies.

2) although data provided by the authors are convincing, an explanation could be addressed of lack of any effect in osteoblasts upon LMNA downregulation. Moreover, do they undergo cellular senescence? Do they secrete IL-6? Are any other cytokines hypersecreted in Lmna-null osteoblasts?

- In the introduction section there are several conceptual mistakes as well as some typing errors. 

Reviewer #3: In the manuscript by Xiong et al., the authors set out to understand the mechanism by which loss-of-function in the Lmna gene leads to osteoporosis. To test this, the authors utilizes two different tissue specific Cre-driver mice to delete lamins A and C in either osteoblast-lineage cells or skeletal muscle cells. Somewhat surprisingly, they find that deletion of Lmna specifically in skeletal muscle has a much more significant effect on bone homeostasis compared to deletion in the actual bone. Mechanistically, the authors identify IL-6 as being a molecule that is secreted from Lmna-deficient skeletal muscle cells, which then increases osteoclastogenesis and leads to increased bone resorption. Lastly, the authors propose that the elevated IL-6 is due to senescence of the skeletal muscle cells and the activation of the senescence -associated secretory phenotype. Overall, the authors provide sound evidence that the effect of lamin mutations on osteoporosis may be due to a secretory phenotype, which is potentially independent of loss of contractile forces on the skeletal system. In particular, the timing of the elevated IL-6 prior to any significant muscle dysfunction is consistent with the model that IL-6 is causing the bone resorption. There is clear importance for this work, as it is still not clear how Lmna mutations lead to disease, however I have some concerns with the interpretation of the data. My specific comments are below.

Major Comments

-While the authors have presented convincing data the Lmna KO -> p16ink4a -> IL-6 -> osteoclastogenesis is a major contributor impaired bone homeostasis, the data to show that this is due to senescence is very limited, and in my opinion, somewhat speculative. I think that the authors would have to provide significantly more data to conclude that loss of lamins A/C leads to senescence of muscle fibers. One of the hallmarks of senescence is loss of proliferation, which is obviously not relevant in post-mitotic muscle fibers. Furthermore, it seems unlikely that in vitro muscle cells that have been differentiated for only 4-5 days, or the 1-month time point in vivo, is sufficient time to undergo senescence. While p16ink4a is associated with senescence, it can be activated by a number of other things, including ROS and DNA damage, which have also been shown to occur in lamin- deficient or -mutant cells (PMID: 25996284, 16772334, 30696354, 31844279) and these two factors may or may not lead to senescence. If the authors want make a stronger case for the myofibers undergoing senescence, I think they need to provide data for other hallmarks of senescence (i.e. increased β-galactosidase activity, loss of lamin B1), otherwise limit the interpretation to suggest that p16ink4a could be elevated for a number or reasons, including cellular senescence.

- I would be cautious with the nomenclature used to describe the phenotypes. In the laminopathy field, it's fairly well accepted that the progeroid-phenotype is very specific to a subset of mutations. I believe that many would argue that full deletion of the Lmna gene do not reflect the progeroid phenotype (PMID: 24947239). Consider limiting the description of the LmnaHSA-cko mouse to a "laminopathy-like phenotype" to avoid specifically comparing it to the progeria-phenotype. 

-The authors must clarify what control was used for the LmnaHSA-cko mice and include the correct nomenclature in the figures. It is now well recognized that just the expression of Cre can have significant effects in straited muscle (PMID: 29858650, 31125255, 23929940) particularly with respect to DNA damage (PMID: 23929941, PMID: 30917606). 

Minor Comments

- There are a few instances where I believe laminopathies is misspelt laminopacies 

-Description of the expression of lamin levels in Ocn-Cre BMSCs and BMMs in confusion. The way it is written now it seems like the BMMs do not have the transgene, but really it's just not being. Correct?

-It's not completely clear why the HSA-Cre was quantified sometimes after 6 months but the Ocn-Cre was not. Is it possible the Ocn-Cre mice would have developed a phenotype eventually?

-The authors need to describe which IL-6 KO strain was used and how the double mutants were generated. There is no mention of this in the methods

Figures

-In some cases the figure legends are quite hard to follow, as the description of multiple panels is combined into one sentence. If possible, consider breaking these up to make it easier to follow.

-Sometimes panel lettering is going across and sometimes is it going down. Again, this just makes the figure hard to follow.

-Figure 5C. I believe this is mislabeled with "CM" when the legends states this is skeletal muscle

---

## [Decision Letter · Decision Letter 2]

11 Apr 2020

Dear Dr Xiong,

Thank you for submitting your revised Research Article entitled "Linking skeletal muscle aging with osteoporosis by Lamin A/C –deficiency" for publication in PLOS Biology. I have now obtained advice from the original reviewers and have discussed their comments with the Academic Editor. 

Based on the reviews, we will probably accept this manuscript for publication, assuming that you will modify the manuscript to address the remaining points raised by reviewer 2. Please also make sure to address the data and other policy-related requests noted at the end of this email.

We expect to receive your revised manuscript within two weeks. Your revisions should address the specific points made by reviewer 2. In addition to the remaining revisions and before we will be able to formally accept your manuscript and consider it "in press", we also need to ensure that your article conforms to our guidelines. A member of our team will be in touch shortly with a set of requests. As we can't proceed until these requirements are met, your swift response will help prevent delays to publication.

*Copyediting*

*Published Peer Review History*

*Early Version*

*Submitting Your Revision*

Sincerely,

Di Jiang

PLOS Biology

ETHICS STATEMENT:

-- Please create a separate Ethics Statement subsection and place it in the beginning of the Methods section. 

-- Please include the full name of the IACUC/ethics committee that reviewed and approved the animal care and use protocol/permit/project license. IMPORTANT: Please also include an approval number.

-- Please include the specific national or international regulations/guidelines to which your animal care and use protocol adhered. Please note that institutional or accreditation organization guidelines (such as AAALAC) do not meet this requirement.

-- Please include information about the form of consent (written/oral) given for research involving human participants. All research involving human participants must have been approved by the authors' Institutional Review Board (IRB) or an equivalent committee, and all clinical investigation must have been conducted according to the principles expressed in the Declaration of Helsinki.

DATA POLICY:

Regardless of the method selected, please ensure that you provide the individual numerical values that underlie the summary data displayed in the following figure panels as they are essential for readers to assess your analysis and to reproduce it: Figures 1BEFHIKL, 2BCEH, 3ACDEFIKG, 4CF, 5CE, 6CD, 7BCDEFGH, 8BDEG, S2BDEHIJKL, S3BCE, S4BCEFH, S6CDEF, S7ABCDE, S8B, S9BCEF, S10BDEHG. NOTE: the numerical data provided should include all replicates AND the way in which the plotted mean and errors were derived (it should not present only the mean/average values).

IMPORTANT: Please also ensure that figure legends in your manuscript include information on where the underlying data can be found, and ensure your supplemental data file/s has a legend. You could write in every relevant legend: e.g., "The underlying data are included in S1 Data" or "The numerical data are included in S1 Data".

Reviewer remarks:

Reviewer #1: The authors addressed all the concerns raised. I would recommend the paper to be accepted for publication.

Reviewer #2: The manuscript is a very important research linking with a nice molecular and genetic approach muscle senescence and osteoporosis. Its interest goes beyond the field of lamin-related mechanisms and can provide new hints for osteoporosis and sarcopenia studies. Further, it provides new ideas for the study of laminopathies.

Minor changes are needed in grammar and style.

Please, add reference to the paper by Rufo et al J Bone Min Res 2011, which is the first study linking muscle disorders to osteoporosis through IL6.

Reviewer #3: The authors did a very nice job responding to the reviewers comments. These additional experiments greatly enhanced the quality of the manuscript.

---

## [Editor Report · Decision Letter 3]

14 May 2020

Dear Dr Xiong,

On behalf of my colleagues and the Academic Editor, Colin Stewart, I am pleased to inform you that we will be delighted to publish your Research Article in PLOS Biology. 

Early Version

PRESS 

Kind regards,

Vita Usova, 

Publishing Editor 

PLOS Biology

on behalf of

Di Jiang,

Associate Editor

PLOS Biology